



# Effect of NO$_X$, O$_3$ and NH$_3$ on sulfur isotope composition during heterogeneous oxidation of SO$_2$: a laboratory investigation

Zhaobing Guo[1], Mingyi Xu[1], Yuxuan He[1], Shuo Gao[1], Chenmin Xu[2], Bin Zhu[3], Qingjun Guo[4], Xiaoyu Shen[1], Shuang Zhao[1], Pengxiang Qiu[1]

[1]Jiangsu Key Laboratory of Atmospheric Environment Monitoring and Pollution Control (AEMPC), Collaborative Innovation Center of Atmospheric Environment and Equipment Technology (CIC-AEET), School of Environmental Science and Engineering, Nanjing University of Information Science and Technology, Nanjing 210044, China

[2]School of Environment, Nanjing Normal University, Nanjing 210023, China

[3]School of Atmospheric Physics, Nanjing University of Information Science and Technology, Nanjing 210044, China

[4]Insititude of Geographic Sciences and Natural Resources Research, Chinese Academy of Sciences, Beijing 100101, China

Correspondence to: Chenmin Xu (chenminxu@njnu.edu.cn) and Pengxiang Qiu (pxqiu@nuist.edu.cn)

**Abstract.** Sulfate aerosol is a major fraction of haze, playing an important role in aerosol formation and aging processes. In order to understand the mechanism of sulfate formations, the characteristics of sulfur isotope composition were determined during different heterogeneous oxidation reactions of sulfur dioxide. Although NH$_3$ was more beneficial to the formation of

sulfate, compared with NO$_X$ and O$_3$, $^{34}$S tended to enrich the lighter sulfur isotopes in the presence of NH$_3$. Furthermore, in consideration of the potential competitive effects of NO$_X$, O$_3$, and NH$_3$ in the heterogeneous oxidation processes, the contributions of each gas were evaluated via Rayleigh distillation model. Notably, NO$_X$ oxidation contributed 67.5±10 % of the whole sulfate production, which is higher than O$_3$ (13.3±10 %), and NH$_3$ oxidation (19.2±10 %) on the basic of the average fractionation factor. The observed δ$^{34}$S values of sulfate aerosols were negatively correlated with sulfur oxidation

ratios, owing to the sulfur isotopic fractionations during the sulfate formation processes. Given the isotope mass balance, the overall δ$^{34}$S$_{sulfate}$ approached the δ$^{34}$S$_{emission}$ as oxidation of SO$_2$ progressed, suggesting that NO$_X$ played a major rather than a sole role in the different heterogeneous oxidation processes of SO$_2$.

**Keywords.** Sulfur isotope, oxidation mechanism, fractionation, Rayleigh distillation model

## 1 Introduction

Haze, a pollution phenomenon that reduces atmospheric visibility to less than 10 km due to the fine particles suspended in the air (Guo et al., 2018; Chen et al., 2019), has aroused widespread concerns in China (Huang et al., 2014; Zhang et al., 2012; DiTucci et al., 2018). The long-term exposure to haze can induce rhinitis and bronchitis, and even lead to lung cancer (Yinon et al., 2017; Wang et al., 2014; Yang et al., 2018). It has been widely recognized that the key factor to the mechanism of haze is the growth of the secondary pollutants and aerosol (Zhang et al., 2019; Zhao et al., 2013; Liu et al., 2012).

Sulfur (S), as an element in sulfate aerosol and various secondary pollutants, is the key to study the formation and evolution of haze. The haze mainly contains sulfur compounds such as H$_2$S, SO$_2$, and SO$_3$, among which sulfur dioxide (SO$_2$)



emission from fossil burning is the main source of anthropogenic sulfate aerosol (Haywood et al., 2000). $SO_2$ can be converted to sulfate via the gas-phase oxidation and heterogeneous reactions (Wang et al., 2019; Kong et al., 2014). Sulfate aerosol not only affects the global climate through direct and indirect radiative forcing (Anderson et al., 2003; Rosenfeld et al., 2014), but also threatens the human respiratory system (Liu et al., 2019). Studying the source and formation mechanism of sulfate aerosol in the atmosphere is of great significance for clarifying the climate change characteristics (Chen et al., 2015). However, the formation mechanism of sulfate is still unclear. Guo et al. (2017) reported that catalytic oxidation of metal ions contributed the most to the formation of sulfate. It was confirmed that $SO_2$, $NO_X$, and $NH_3$ synergistically accelerated the formation of sulfate under high humidity (Wang et al., 2016). Li et al. (2018) demonstrated that the oxidation of S(IV) in haze in China was driven by the $HONO/NO_2^-$ generated due to the consumption of $NO_2$ on the surface of aerosols. Besides, aqueous $NO_2$ serves as the dominant oxidant of $SO_2$ at highly elevated $NO_X$ levels (Xue et al., 2019). Moreover, the synergistic effect between $NO_2$ and $SO_2$ on the surface of mineral promotes the conversion of $SO_2$ to sulfate. Therefore, the oxidation processes of $SO_2$ cannot be summarized by a simple oxidation mechanism.

The heterogeneous oxidation is the core process of the formation of secondary aerosol, having a guiding role in revealing the formation mechanism of the compound pollution (Lu et al., 2018). The measured sulfur isotopic fractionation showed that about -9 ‰ is for homogeneous oxidation of $SO_2$ and up to +16.5 ‰ is for heterogeneous oxidation of $SO_2$ (Chen et al., 2017). In addition, the pathways of $SO_2$ oxidation in aqueous-phase systems include reactions with $O_3$, $H_2O_2$, $NO_2$, and by $O_2$ via catalyst (Hung et al., 2015). The formation of sulfate was demonstrated that it was mainly affected by a synergistic effect between $NO_X$ and $SO_2$ (Gao et al., 2020). Sulfate formations measured during autumn were mainly related to excessive $O_2$ with $Fe^{3+}$ as catalyst (Guo et al., 2014). Therefore gaseous oxides are the key oxidation pathways in the formation of sulfate during the heterogeneous oxidation process (Chen et al., 2017).

The formation process of the secondary sulfate and sources of sulfur in the atmosphere can be investigated by sulfur isotope ratios (Han et al., 2017), as the characteristics of different sulfur sources can be represented by sulfur isotopic signatures ($^{32}S$ and $^{34}S$) (Winterholler et al., 2008). Besides, sulfur isotopes also exhibit distinctive isotope fractionations for different oxidation processes of $SO_2$ (Han et al., 2016a), which can be applied in the study of formation processes of sulfate. The enrichment of heavy sulfur isotopes in sulfate may be caused by heterogeneous oxidation of $SO_2$, whereas light sulfur isotopes in sulfate may be attributed to homogeneous oxidation (Chen et al., 2017). Nevertheless, isotope fractionations can only be used to roughly distinguish between the heterogeneous and homogeneous oxidation. The specific oxidation path of $SO_2$ cannot be discriminated due to the similarity of the sulfur isotope fractionation under different conditions during the oxidation processes of $SO_2$.

Some investigations about sulfur isotope composition and fractionation have been performed for understanding the formation pathways of sulfate aerosols (Harris et al., 2013b; Yang et al., 2018; Chen et al., 2017). However, most of these observational and modeling studies investigate the sulfur isotope composition in real atmosphere without physical boundaries (Han et al., 2016b; Li et al., 2020). Yet to date, a few experiments were performed in limited physical boundaries to explore the mechanism of sulfur isotope fractionation on the microscale (Harris et al., 2012a; Harris et al., 2012b). To our best knowledge, the effect of $NO_X$, $O_3$ and $NH_3$ on sulfur isotope composition during heterogeneous oxidation of $SO_2$ have not been determined experimentally. Herein, for the first time, the several $SO_2$ oxidation processes with different chemical condition ($NO_X$, $O_3$ and $NH_3$) are conducted in laboratory to gain insight into the sulfur isotope fractionation. Furthermore,



the sulfur isotopic fractionations were investigated with the Rayleigh distillation model to understand the relative contribution of each $SO_2$ oxidation pathway. It may not only provide a theoretical basis for the causes of subsequent sulfates, but also be crucial for improving the air quality and studying the regional climate change.

## 2 Materials and Methods

### 2.1 Material and methods

Sampling site was located on the roof of the library in Nanjing University of Information Science & Technology (32.1°N, 118.5°E). $PM_{2.5}$ samples were collected by using a high volume sampler (TH-1000H, Tianhong Co., Wuhan) with a flow rate of 1.05 $m^3$ $min^{-1}$ from 9 am to 9 pm per day from 26th Feb. 2016 to 6th Apr. 2016.

Hematite (α-$Fe_2O_3$) in the experiments was prepared according to the previous studies (Legodi et al., 2007; Fu et al., 2006). A plate with evenly dispersed α-$Fe_2O_3$ powder was loaded into the experimental apparatus. On the basis of $SO_2$-Ar, different proportions of $NO_X$ ($O_3$ or $NH_3$) were added to the reactor, combined with/without $O_2$ and/or light. The $NO_X$ was composed of $NO_2$ and NO with volume ratio of 2:1. The flow rates of Ar and $SO_2$ were 95 and 2 mL $min^{-1}$ respectively. The flow rates of $NO_X$ ($O_3$ or $NH_3$) varied from 2 to 16 mL $min^{-1}$ depending on the ratio of $SO_2$ to $NO_X$ ($O_3$ or $NH_3$). The wavelength of ultraviolet (UV) light is 303 nm, and the light intensity is 25 μW $cm^{-2}$. All of the experiments were conducted for 2 h at a temperature of 298 K with a relative humidity of nearly 40 %. The values of temperature and humidity in the experiments were set similarly to those in the air during this period.

The obtained samples were soaked in 50 mL of Milli-Q water and sonicated for 30 minutes to extract sulfate. The samples were centrifuged to separate the sulfate supernatant. The dissolved sulfate in the supernatant was precipitated as $BaSO_4$ by adding 1 mol $L^{-1}$ $BaCl_2$ solution. The $BaSO_4$ precipitates were separated with 0.22 μm acetate membrane and rinsed with 150 mL Milli-Q water to remove $Cl^-$. The $BaSO_4$ powder was calcined at 1123 K in a muffle furnace for 2 h to obtain the final pure $BaSO_4$ sample.

### 2.2 Sulfur stable isotope determination

The $δ^{34}S$ value was determined by using isotope mass spectrometer (IRMS, Delta V Plus, Finningan) and Elemental analyzer (EA, Flash 2000, Thermo). $BaSO_4$ was used as an analytical sample of sulfur isotope composition. The result was with respect to international standard V-CDT, and the accuracy was better than ±0.2 %.

## 3 Results and discussion

First all, for confirming the sulfur isotope composition and fractionation, $PM_{2.5}$ samples from 26th Feb. to 6th Apr. were collected and the $δ^{34}S$ values of them were measured. $δ^{34}S$ value may change when $SO_2$ is oxidized into the sulfate in the atmosphere via different oxidation pathways. $δ^{34}S$ aerosol and $SO_2$ Oxidation Ratio (SOR = $SO_4^{2-}$/($SO_4^{2-}$ +$SO_2$)) calculated throughout the sample period were shown in Fig. 1. It can be found that the $δ^{34}S_{aerosol}$ values showed a ~3.9 ‰ variation, and displayed a negative correlation with SOR. The variation of $δ^{34}S_{aerosol}$ values was attributed to the isotope fractionation during the oxidation processes of $SO_2$ (Li et al., 2020). To shed light on the mechanism of $SO_2$ oxidation in sulfur isotope fractionation and sulfate formation, the $SO_2$ oxidation processes on the surface of α-$Fe_2O_3$ in the presence of $NO_X$, $O_3$, and $NH_3$ were carried out in laboratory.



$NO_X$ is the most important oxidant during the heterogeneous oxidation of $SO_2$ taking the impact of ion strength into account. As shown in Fig. 2, the yield of $SO_4^{2-}$ ranged from 0.0097 to 0.7795 g and the values of $\delta^{34}S$ were 2.9–4.8 ‰. It was noteworthy that there was an obvious discrepancy between the yield of $SO_4^{2-}$ and the sulfur isotope values. Few sulfates were formed via the heterogeneous reaction between $NO_X$ and $SO_2$ on the surface of mineral in the dark, whereas the formation of sulfate was enhanced under light, suggesting that UV light could promote the oxidation of $SO_2$. Besides, in the presence of $O_2$, $NO_X$ and mineral oxides could act as catalysts to increase the conversion rate of $SO_2$ on the surface of mineral oxides (Gao et al., 2020). In addition, the increase in the amount of $NO_X$ was another key factor that led to the acceleration of sulfate formation (Cheng et al., 2016). When $NO_X$ and $SO_2$ coexisted, the content of sulfite on the surface of all oxides was reduced significantly (Pan et al., 2019). Simultaneously, HONO was formed by $NO_2$ and subsequent hydrolysis in thin films of water coating boundary layer surfaces according to reaction (R1) (Kebede et al., 2016). The release of HONO may help to sustain the efficient sulfate production and droplet acidity. Moreover, the co-adsorption of the oxidant $N_2O_4$ formed from nitrate under the action of S(IV) further led to the formation of sulfate (Cheng et al., 2016).

$$NO_X + H_2O \rightarrow HONO + HNO_3, \tag{R1}$$

$$NO_2 \xrightarrow{\text{S(IV)}} N_2O_4, \tag{R2}$$

$$N_2O_4 + SO_3^{2-} \rightarrow SO_4^{2-} + NO_2^-, \tag{R3}$$

The different proportion of $SO_2$ and $NO_X$ made a great growth of the sulfur isotope values, which suggested that the $\delta^{34}S$ values were largely dependent on the concentration of $NO_X$. The sulfur isotope values all reached the maximum when $SO_2$ : $NO_X$ was 1:8 under different conditions, indicating a significant sulfur isotopic fractionation effects. It was observed that the $\delta^{34}S$ values were relatively lower in the presence of only $O_2$. It was well-known that the reaction between $SO_2$ and $O_2$ was conducted via radical chain reactions in the presence of Fe(III) (Hung et al., 2015), which favored the enrichment of lighter isotopes (Han et al., 2016b). Therefore, under dark conditions, the sulfur isotope value was mainly affected by oxygen and the catalytic action of Fe(III), resulting in the enrichment of lighter sulfur isotopes (Han et al., 2016a). In addition, the $\delta^{34}S$ values of vehicle exhaust from which $NO_X$ emissions were primarily derived were much higher compared to those of coal combustion (Grewling et al., 2019; Guo et al., 2016), indicating that $^{34}S$ tended to enrich higher sulfur isotopes in the presence of $NO_X$.

$O_3$ also exerted an influence on the formation of sulfate and sulfur isotope values. The yield of $SO_4^{2-}$ ranged from 0.0081 to 0.6712 g with the $\delta^{34}S$ values of 1.6–2.9 ‰ (Fig. 3). The rapid growth sulfate yields indicated that the oxidation of $SO_2$ was sensitive to the concentration of ozone. Ozone, as a very efficient oxidant, could react with the sulfite to release oxygen, promoting the subsequent oxidation of $SO_2$, which was described as:

$$SO_3^{2-} + O_3 \rightarrow SO_4^{2-} + O_2, \tag{R4}$$

$$HSO_3^- + O_3 \rightarrow HSO_4^- + O_2, \tag{R5}$$

Whether the oxidation processes were on the surface of $Fe_2O_3$ mineral dust or not, the coexistence of $O_3$ can convert $SO_2$ to sulfate (Gao et al., 2020). The highest sulfate production obtained at a ratio of 1:12 may be attributed to the higher photochemical activities and ozone concentration (Kong et al., 2019). Besides, irradiation had been demonstrated to prevent surface saturation for ozone uptake on mineral (Nicoals et al., 2019). These results suggested that experimental conditions, such as ozone and irradiation, changed the quantity of $SO_2$ taken up and oxidised. Under dark conditions, photolysis of $O_3$ were negligible, thus surface reactions will be solely responsible for sulfate production (Harris et al., 2013a). Moreover, the



photolysis of ozone under UV radiation formed electronically excited $O(^1D)$, and its subsequent reaction with water vapor could generate a mass of hydroxyl radicals (Ran et al., 2014; Cheng et al., 2016). As adsorption sites for water, surface hydroxyls were the principal reactive sites on metal oxides. In turn, the adsorbed water was either dissociated into more hydroxyls at oxygen vacancies or hydrogen-bonded to surface O-H groups, which was in favor of the heterogeneous oxidation of $SO_2$ (Wang et al., 2019).

$$O_3 + h\upsilon \rightarrow O(^1D) + O_2, \tag{R6}$$

$$O(^1D) + H_2O \rightarrow 2\bullet OH, \tag{R7}$$

$$\bullet OH + SO_2 + M \rightarrow HO_2 + SO_4^{2-}, \tag{R8}$$

It can be observed from Fig. 3b that the elevation of $\delta^{34}S$ values was related to the concentration of $O_3$. The oxidation by $O_3$ dominated in the reaction and favored heavy sulfur isotopes (Han et al., 2016b). The high $\delta^{34}S$ values, especially the

highest value obtained at the ratio of 1:8, may be in relation to $\bullet OH$ which promoted the enrichment of heavy sulfur isotopes (Harris et al., 2012b). The values of sulfur isotope in the presence of $O_2$ were relatively lower than that under the condition of both $O_2$ and light, indicating that the synergistic effect of $O_2$ and light facilitated the enrichment of higher sulfur isotope values. In addition, uptake and decomposition of ozone under irradiation increased the basicity of the surface, which was conducive to enrich heavy sulfur isotopes (Hanisch et al., 2003). Harris et al. (2012) confirmed the hypothesis that

equilibration to higher pH increased fractionation.

NH$_3$, as another one of the main pollutants, played an important role in the formation of atmospheric sulfate. It can be observed that the yield of $SO_4^{2-}$ ranged from 0.0237 to 0.9469 g with the $\delta^{34}S$ values of 0.8–4.3 ‰ (Fig. 4). Acidic aerosols can react with gaseous ammonia to form completely or partially neutralized ammonium salts (Donaldson et al., 2010). The extent of aerosol neutralization was determined principally by the ambient concentration of $NH_3$ and the oxidation rate of

$SO_2$ (Kong et al., 2019). When ammonia was in excess, sulfate aerosol should be mainly presented as ammonium sulfate (Silvern et al., 2017). Herein, the results showed that the reaction of $NH_3$ with acid may lead to an increase in the formation of sulfate. In addition, surface Lewis basicity might be provided by $NH_3$ for $SO_2$ absorption on the mineral, increasing the amount of condensed water on the secondary aerosols and enhancing the formation of sulfate (Chu et al., 2016). Moreover, the oxygen vacancies in $\alpha$-$Fe_2O_3$ may lead to the formation of sulfate on $\alpha$-$Fe_2O_3$ (Wang et al., 2019).

Of note, the effect of $NH_3$ on the sulfur isotope composition was not apparent with the relatively stable overall trend. Under only-light, $NH_3$, which increased the alkalinity by producing $OH^-$ from hydrolysis, dominated in the reaction, leading to an increase of $\delta^{34}S$ values (Jiang et al., 2017). $O_2$ with $Fe^{3+}$ as catalyst dominant in the presence of only-oxygen favored light sulfur isotopes, which was consistent with above results. The $\delta^{34}S$ values of sulfate increased with the increases of sulfate concentrations under the combined oxygen with light (Doi et al., 2004). Therefore, we inferred that $O_2$ and light had a

synergistic effect on the sulfur isotope compositions in the presence of $NH_3$. Compared with the $\delta^{34}S$ values in the presence of $NO_X$ above, S under $NH_3$ was more inclined to enrich the lighter sulfur isotopes. The $\delta^{34}S$ values from main biogenic source of $NH_3$ were on the low side, indicating that the effects of $NH_3$ on sulfur isotopic compositions were relatively mild (Han et al., 2016a; Grewling et al., 2019). Simultaneously, there was only a slight fluctuation of the $\delta^{34}S$ values with the rapidly increasing of the sulfate production in the presence of $O_2$ and light. Hence, the existence of $NH_3$ increased the

conversion rate of $SO_2$ and the sulfate production with little effects on the isotope fractionation.

To make clear the relative contribution of $NO_X$, $O_3$, and $NH_3$, we investigated the isotope fractionation of sulfate. The





oxidation of $SO_2$ to sulfate caused the fractionation of isotope ratios as long as the reaction is not complete. Isotope fractionation is divided into equilibrium and kinetic fractionation. The coefficient α considered as the kinetic rate constants ratio can represent the fractionation effects. When the reactant is presented as an infinite reservoir and not affected by the

reaction, $α_{34}$ can be calculated from the isotopic compositions of reactants and products (Harris et al., 2012a):

$$\alpha_{34} = \frac{R_{products}}{R_{reactants}},$$  (1)

Among them, $R = ^{34}S/^{32}S$. Thus, $\alpha > 1$ indicates that the reactions are more inclined to enrich heavy isotopes.

The $\delta^{34}S$ value of $SO_2$ sources ($\delta^{34}S_{emission}$) strongly depends on the origin of $SO_2$, thus it is difficult to constrain (Li et al., 2018). On the basis of the literature, $+2.7\pm2.0$ ‰ is used as the $\delta^{34}S_{emission}$ value during our sampling period (Li et al.,

2020). Therefore, we used a Rayleigh distillation model and the isotopic enrichment factor ($\varepsilon = (\alpha-1)*1000$‰) to explain the discrepancy between $\delta^{34}S$ and $\delta^{34}S_{emission}$ values and quantify the contributions of each oxidant on the basis of the above results in the experiment. The fractionation factor is found from the Rayleigh equation describing the $\delta^{34}S_{sulfate}$ with respect to the fraction of oxidized $SO_2$ (Harris et al., 2012a):

$$\alpha_{34} = \frac{\ln(1-(\frac{R_p}{R_i})(1-f))}{\ln f},$$  (2)

Where $R_p$ and $R_i$ are the ratios of $^{34}S/^{32}S$ for the product sulfate and the initial $SO_2$ gas respectively and f (1-SOR) is the fraction of remaining $SO_2$.

The measured $\varepsilon_{obs}$ values as a result of mixing oxidation pathways of $SO_2 + NO_X + O_3 + NH_3$ were $+1.3\pm1.4$ ‰ by simulations (Fig. 5). $NO_X$ oxidation enriched $^{34}S$ in the product sulfate with an enrichment factor ($\varepsilon_{NOX}$) of $+2.1$ ‰, and oxidation by $O_3$ pathway depleted $^{34}S$ ($\varepsilon_{O_3}=+0.2$ ‰) in the product sulfate, and oxidation by $NH_3$ pathway enriched $^{34}S$ in the

product sulfate with a $\varepsilon_{NH3}$ value of -0.6 ‰. Considering the isotope mass balance, the overall $\varepsilon_{obs}$ value (+1.3 ‰) fell in between NOx, $O_3$, and $NH_3$ values and approached $\delta^{34}S_{emission}$ as oxidation of $SO_2$ progressed, indicating that $NO_X$, $O_3$, and $NH_3$ all had a certain influence on the heterogeneous oxidation of $SO_2$.

$$\varepsilon_{obs} = \varepsilon_{NOx} * f_{NOx} + \varepsilon_{O_3} * f_{O_3} + \varepsilon_{NH_3} * f_{NH_3},$$  (3)

In which $\varepsilon$ and f are the enrichment factor and the contribution of different pathways respectively, and $f_{NOx} + f_{O_3} + f_{NH_3} =$

200 1.

Using the Eq. 3, we determined the overall contributions from NOx, $O_3$, and $NH_3$ pathways were $67.5\pm10$ %, $13.3\pm10$ %, $19.2\pm10$ %, respectively. During the period of collecting $PM_{2.5}$, the average concentration of $NO_X$ and $O_3$ showed a negative correlation, and the overall concentration of $NO_X$ in the atmosphere was lower than that of $O_3$. However, $NO_X$ accounted for a larger proportion in sulfur isotope fractionation due to its tendency to enrich heavier sulfur, suggesting

the $NO_X$ pathway have played a more important role during the oxidation of $SO_2$.

## 4 Conclusion

$NO_X$ exerted a driving influence on the enrichment of heavy sulfur isotopes. Hydroxyl radical formed by the photolysis



of $O_3$ favored the enrichment of relatively heavy sulfur isotopes. Although $NH_3$ was more conducive to the formation of sulfate compared with $NO_X$ and $O_3$, there existed lighter sulfur isotopes with a slight effect on the isotope fractionation in the presence of $NH_3$. The evaluated contributions of $NO_X$, $O_3$, and $NH_3$ oxidations were 67.5±10 %, 13.3±10 %, and 19.2±10 % respectively via the Rayleigh distillation model and average fractionation factor. On the basis of the isotope mass balance, we concluded that $NO_X$ pathway was a dominating but not sole pathway during different heterogeneous oxidation processes of $SO_2$, which laid the foundation for the research of the mechanism of sulfate formation.

**Author contribution.**

ZBG designed the methodology, administrated the project and wrote the original draft. MYX performed model simulations and analyzed the data. YXH conducted the investigation process. SG provided the study materials. XYS performed the data collection. SZ provided the laboratory samples. BZ, QJG and CMX wrote the review and commentary. PXQ reviewed and revised the paper.

**Acknowledgment**

We gratefully acknowledge the financial supports from the National Natural Science Foundation of China (Nos. 41873016, 51908293, 51908294 and 41625006), National key R&D program of China (91544229-02), Jiangsu Province 333 Talent Project (BRA2018033) and the Natural Science Foundation of Jiangsu Province (BK20190718).

**Competing interests.** The authors declare that they have no conflict of interest.

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

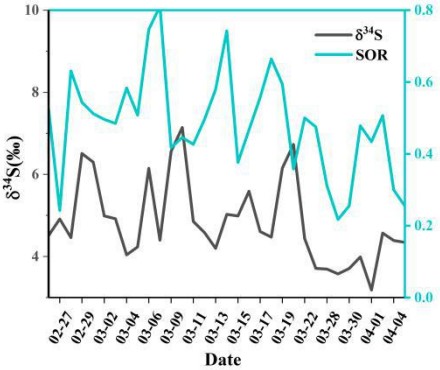

**Figure 1:** $\delta^{34}S_{aerosol}$ (black), calculated sulfur oxidation ratio (SOR, green) throughout the sample period.

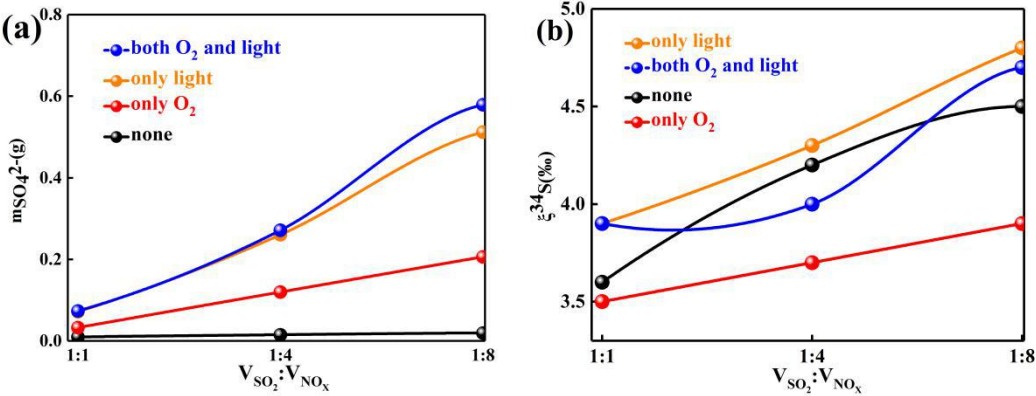

**Figure 2:** Effect of mixed gas of $SO_2$ and $NO_X$ on (a) sulfate production and (b) sulfur isotope value under different reaction conditions.





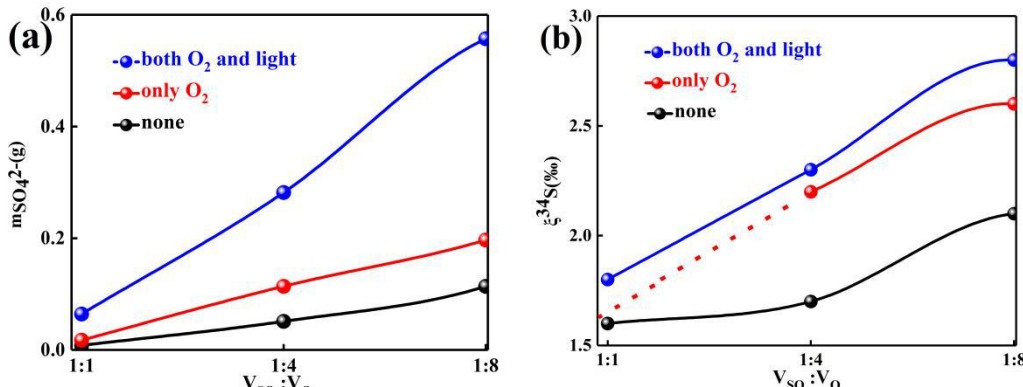

**Figure 3:** Effect of mixed gas of $SO_2$ and $O_3$ on (a) sulfate production and (b) sulfur isotope values under different reaction conditions. (The dotted line represents the trend as it should be).

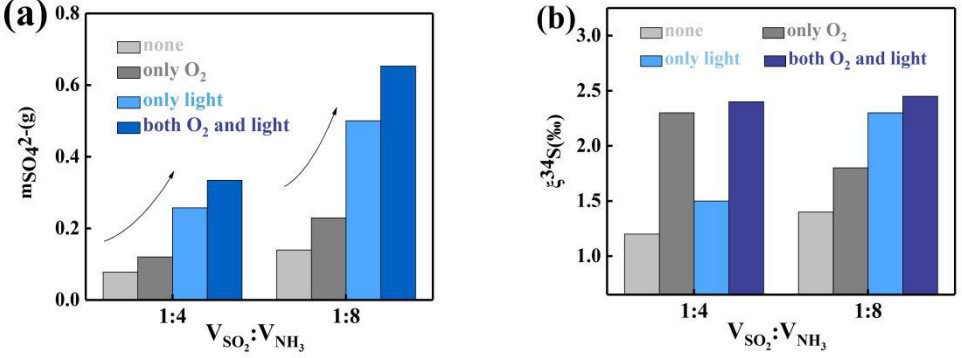

**Figure 4:** Effect of mixed gas of $SO_2$ and $NH_3$ on (a) sulfate production and (b) sulfur isotope values under different reaction conditions.

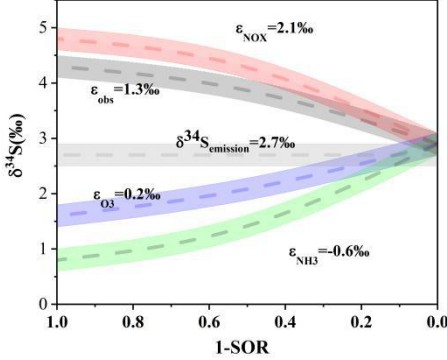





**Figure 5:** Rayleigh distillation model of sulfate production. Grey bar indicates the $\delta^{34}S_{emission}$ (+2.7±2 ‰) in Nanjing. Dashed lines with shaded areas are $\delta^{34}S_{sulfate}$ values: red line indicates the $\delta^{34}S_{sulfate}$ when $SO_2$ is oxidized solely by $NO_X$, blue line indicates the $\delta^{34}S_{sulfate}$ when $SO_2$ is oxidized solely by $O_3$, green line indicates the $\delta^{34}S_{sulfate}$ when $SO_2$ is oxidized solely by $NH_3$, and dark grey line is the $\varepsilon_{obs}$ of +1.3 ‰.