# Peer review of "Effect of NOx, O3 and NH3 on sulfur isotope composition during heterogeneous oxidation of SO2: a laboratory investigation"

_Atmospheric Chemistry and Physics, 2020_

## Referee Comment (RC1) · Anonymous Referee #1 · 24 Jul 2020

Major comments

I think this study adds information on the sulfur isotopic fractionation ratio ($\delta$S34) from several heterogeneous SO2 oxidation pathways involving hematite with NOx, ozone, and NH3. However, I do not think that the results presented here allow the authors to draw conclusions about the relative importance of different pathways in real field observations. This is my largest concern. The authors only include the three heterogeneous pathways they measured in this study's laboratory experiments when interpreting observed $\delta$S34 using the Rayleigh distillation model. However, there are also other aqueous-phase and gas-phase reactions with different fractionation factors (e.g.

[Figure]

Harris et al., 2012a, b; who also show it is not possible to distinguish between some pathways based on S34 alone). For example, Shao et al. (2019) found heterogeneous oxidation of SO2 on all aerosols was only 20% in wintertime in China, with alkaline dust (for which hematite is possibly an analog) contributing a total of only 9%. Even without considering additional reactions which would create a very underconstrained problem with no solution, the authors are solving for three unknowns (the relative importance of their three studied reactions) using only two equations (equation 3 and line 199). How is this possible? I am a bit confused that the authors do not show or use the concentrations of potential oxidants measured at the field site to address the relative importance of different pathways, although they are vaguely referred to in lines 202-203.

I think the authors should also explain in more detail why hematite is chosen. Is it a true analog of mineral dust? How do the reactions involving hematite in this paper relate to the aqueous-phase transition metal ion catalyzed SO2 oxidation discussed at length in the literature? Is there aerosol water present at RH=40% and thus the possibility of aqueous-phase oxidation reactions in hematite leachate? How does this study of fractionation in reactions with only iron-containing hematite relate to/inform interpretation of previous results for actual desert dust (e.g., Harris et al., 2012b)? I think these questions should be addressed explicitly in the paper.

General comments

Overall, I find the explication of the laboratory experimental results confusing as it is difficult to determine what has been found in previous studies vs. what is being interpreted from the results of this study, in light of the literature. For each of the previous studies cited during the explanation of your laboratory results, I would like more information so I know how similar their experimental set-up was and am able to better interpret how your work compares. I will give examples in the line-by-line comments.

I think the focus of the paper should be made more clear - if the purpose is specifically

to address sulfate formation during pollution / haze events, specific seasons and/or in specific regions, or if these results would be applicable generally. Some line-by-line comments below also address this.

Yields of sulfate are currently given in absolute grams – it would be helpful for comparison purposes to also give in terms of a relative fraction of product vs. (SO2) reactant loss.

Line-by-line comments

- Line 37: "the formation mechanism of sulfate is still unclear": is this globally, during pollution/extreme haze events, or in China specifically? Given that there is no discussion of, e.g., DMS oxidation, or oxidations by HOCl/HOBr (Q. Chen et al. (2017), GRL), I think the focus in the introduction should be narrowed appropriately and it would make more clear why which studies are cited in this introduction.

- Line 39: Li et al. (2018): there are two citations that are Li et al. (2018), please specify which one

- Line 42: please add a citation to this sentence ("the synergistic effect. . . promotes the conversion of SO2 to sulfate") or make more clear if it is related to previous citations in this paragraph.

- Line 45-46: "measured sulfur isotopic fractionation" – could you please specify which isotopic ratio this paper is studying (e.g., is it $\delta$S34 or something else)?

- Line 51: there are two Chen et al. (2017) papers in the references, please specify which one

- Line 59-60: please add a citation to this sentence (e.g., Harris et al., 2012b)

- Line 70: I would reword "the causes of subsequent sulfates" to something like "the subsequent formation of sulfate"

- Line 83-84: could you state the average temperature and RH at the sample site? I

don't know what "similarly" means in this context.

- Section 2.1: add a sentence or two to very clearly distinguish between the field sampling vs. the laboratory experiments, e.g., a beginning sentence for each of the first two paragraphs of this section. In fact, I would move the sentence in lines 95-96 in Section 3 to the first paragraph, and the sentence in line 100-102 to the 2nd paragraph.

Figures:

- All figures: it would be very helpful if Figure 2a,3a,4a and 2b,3b,4b had the same y-axis range. It makes it harder to compare across oxidants otherwise. Is each point based on a single measurement, or are multiple samples taken? If the latter, the standard deviation/error should be shown. If not, I think it would be helpful to add either perpendicular lines on the points or shading to indicate the accuracy of the measurements, as $\pm 0.2$ per mil is as large as some of the changes in $\delta 34S$ caused by changing the volume ratios of oxidants. I believe the y-axis label should be changed from to $\delta 34S$.

- Figures 2 and 3: what kind of curve-fitting is being done here? Splines? A polynomial function? I think the nonlinear curves mislead the reader to think that there is a change in the isotopic fractionation between measured values that we cannot know for sure from the experiments performed here alone (e.g., the "both O2+light" blue line in figure 2b between 1:1 and 1:4 ratio makes it seem like it is decreasing, but that is a product of the nonlinear curve-fitting being done with only 3 points. maybe it's actually flat or slightly increasing if it was a linear trend instead). I would either remove the lines entirely and only show points, or simply draw lines between each set of points as this is the simplest type of fitting to do and there is such limited data to justify a polynomial fit.

- Figure 3 caption: "(The dotted line represents the trend as it should be)." What does this mean? This is not explained in the text. I think the dotted line should be removed unless it can be explained why this would be known from theory.

- Figure 4: I would make this a line graph instead of a bar chart, I am not sure why it is different from Figures 2 and 3. Also, I would remove the arrows from Figure 4a, I do know what this adds. Explain why you did not do a 1:1 ratio experiment (e.g., it requires ammonia in excess).

NOx results description:

- Lines 103 – 104 (First sentence of this paragraph): To me, the way this is currently worded makes it unclear whether this is something known before or based specifically on the results of this study and what is shown in Figures 2 to 4. What do you mean by taking ion strength into account? How is that accounted for in the presentation of Figures 2-4?

- Lines 109-110: "In addition, the increase in the amount of NOx was another key factor that led to the acceleration of sulfate formation (Cheng et al., 2016)." It is not clear whether this refers to results from your study, Cheng et al. (2016), or both. Please revise to clarify.

- Lines 123-124: "Therefore, under dark conditions, the sulfur isotope value was mainly affected by oxygen and the catalytic action of Fe(III), resulting in the enrichment of lighter sulfur isotopes (Han et al., 2016a)." Similar to the above comments. Please clarify if this is the case for both your experiments and previous work; if not, please state that explicitly.

- Line 149-151: I think Harris et al. (2012a) should be cited here, which is about OH in the gas-phase. I am not sure though how OH in the gas phase would relate to heterogeneous oxidation here. If I am mistaken, more explanation is needed of how Harris et al. (2012b) relates to your study and OH.

- Line 154: specify which Harris et al. 2012 paper

Ozone results description:

- Reactions R4 & R5: the sentence before says, "which was described as". Please give

the citation where these reactions were described.

- Line 136-137: what mineral did Nicoals et al. (2019) use? Would it behave the same as the hematite used in your study?

- Lines 138-139: "Under dark conditions, photolysis of O3 were negligible, thus surface reactions will be solely responsible for sulfate production (Harris et al., 2013a)." I find this sentence confusing. I think I understand what you mean, but please restate so it is clear that during your study and in general, there is no photolysis of ozone in the dark, so any sulfate production during the dark does not involve the ozone photolysis pathway. I think it would make more sense to introduce the ozone photolysis pathway (lines 140-147) first and then interpret your experimental results in the light and dark.

- Line 153-154: "In addition, uptake and decomposition of ozone under irradiation increased the basicity of the surface, which was conducive to enrich heavy sulfur isotopes (Hanisch et al., 2003)." Again, reword to clarify what you are assuming to apply to your study vs. what you actually can determine from your results. As an example, something like this would help clarify for me as a reader: "It is possible that the uptake and decomposition of ozone under irradiation in our study would increase the basicity of the surface, which Hanisch et al. (2003) found to increase enrichment of heavy sulfur isotopes and is consistent with the increased enrichment in our irradiated ozone experiments."

NH3 results description (lines 156 – 175)

- Line 158-160: "The extent of aerosol neutralization was determined principally by the ambient concentration of NH3 and the oxidation rate of SO2 (Kong et al., 2019)." I don't know if this sentence is needed. In your study, you are not determining the aerosol neutralization, correct? To me, it sounds like it is referencing the methods of the previous study (Donaldson et al., 2010), which described the neuralization of the ammonium salts. I am not sure if this is what you meant.

- Line 163-164: "Moreover, the oxygen vacancies in $\alpha$-Fe2O3 may lead to the formation of sulfate on $\alpha$-Fe2O3 (Wang et al., 2019)." Is this only for the ammonia experiments specifically, or would this statement actually apply to all of your laboratory experiments? Please explain.

- line 166-167 " Under only-light, NH3, which increased the alkalinity by producing OH- from hydrolysis, dominated in the reaction, leading to an increase of $\delta$34S values (Jiang et al., 2017)." : Please separate out which part of this sentence is based on the results found here vs. the part that is coming from Jiang et al. (2017). My guess would be that the $\delta$34S part is the results from this study, and the alkalinity by producing OH from hydrolysis is from Jiang et al. (2017), but it is not obvious to me as a reader.

- Line 167-168: "O2 with Fe3+ as catalyst dominant in the presence of only-oxygen favored light sulfur isotopes, which was consistent with above results." Are the "above results" from Jiang et al. (2017), or the $\delta$34S results found in your laboratory experiment? Can you please explain more clearly why $\delta$34S decreases during only-O2 experiments when ammonia is increased but not when ozone or NOx are increased? How does ammonia affect the transition-metal-ion catalyzed pathway?

- Line 168-169: "The $\delta$34S values of sulfate increased with the increases of sulfate concentrations under the combined oxygen with light (Doi et al., 2004)." Is this the results of Doi et al. (2004) for ammonia experiments? This seems to conflict with your results, which show in Fig. 4b that $\delta$34S stays the same under the combined oxygen with light for higher ammonia (which caused higher sulfate production in figure 4a).

- Line 169-170: "Therefore, we inferred that O2 and light had a synergistic effect on the sulfur isotope compositions in the presence of NH3." I am not sure how this relates to the previous sentence citing Doi et al. (2004). To me it seems that the effects cancel each other out: under increasing ammonia, the "only O2" experiment has decreased $\delta$34S while the "only light" experiment has increased $\delta$34S and the "both O2 + light" experiment has nearly constant $\delta$34S. Perhaps a different word besides "synergistic"

[Figure]

would be more specific.

- Line 171-173: "The $\delta34S$ values from main biogenic source of NH3 were on the low side, indicating that the effects of NH3 on sulfur isotopic compositions were relatively mild (Han et al., 2016a; Grewling et al., 2019)." Again, I do not know what part of this sentence is based on the results of your laboratory experiments here vs. these previous studies that are cited. What does the biogenic nature of NH3 have to do with your experiments?

Rayleigh distillation and enrichment, lines 176-205

- Line 181 (equation 1): It is not clear to me if this equation is needed as it seems it is not used and instead equation 2 (line 289) is. If it is used, can it be made more clear how and why the two equations are presented?

- Line 193 and Figure 5: what does "simulations" mean exactly? What data is specifically being plotted here? What do the shaded regions around each line mean? Please add that to the figure caption. A standard deviation is given for the emissions, but not the other lines.

- Equation 3, Lines 199-202: Please explain how three unknowns are solved with only two equations.

Citations referenced here that are not in the original paper:

Shao et al. 2019: https://www.atmos-chem-phys.net/19/6107/2019/acp-19-6107-2019.pdf

Q. Chen et al. 2017: https://agupubs.onlinelibrary.wiley.com/doi/pdf/10.1002/2017GL073812

---

## Referee Comment (RC2) · Anonymous Referee #2 · 2 Oct 2020

I have read this manuscript, and I found that I mostly agree with previous comments from referee #1. This study provides sulfur isotopic fractionation for sulfate formation from SO2 in the presence of NOx, O3, and NH3. Although these experimental results show some interesting phenomena, I do not think that these results lead to the conclusions drawn by the authors. Note that SO2 has two oxygen atoms and SO42- has four oxygen atoms, thus we have to think the origins of oxygen atoms in sulfate formation. The effect of NH3 for sulfate formation is interesting, because the presence of NH3 may change pH in liquid and promote the pH-dependent process such as O3, TMI, NO2. Unfortunately, I found a lack of this viewpoint throughout this manuscript.
The most important concern related to this experiment is what oxidation processes were included in each experimental system. Previous experimental results by Harris et al. showed the S isotope fractionations for gas-phase oxidation (i.e. SO2+OH) and aqueous oxidations by O3, H2O2, and O2 catalyzed by TMI. They also reported fractionation in SO2 oxidation on the dust surface. Compared to these results, this manuscript provides S isotopic fractionation for sulfate formation with different conditions, but I do not understand which oxidation processes were occurred in each system. Simply speaking, I do not understand which oxidation processes, which is so confusing. Thus, I do not agree that this experiment can directly be applied for the interpretation of observational data sets.

The 2nd important concern is the conclusions of this study that NOx played a major in the different heterogeneous oxidation process of SO2, which cannot be lead by these experimental results and interpretation. Particularly, in eq (3), authors hypothesized that sulfate is only formed via three pathways of SO2 + NOx, O3, and NH3, but this is not true (as mentioned above). Thus, the conclusion lead by this calculation is not appropriate. These comments are almost the same as referee #1 of "how is this possible?".

Overall, I think this manuscript should be reconsidered. Detailed comments from referee #1 were very helpful and I do not have additional comments.

---

## Referee Comment (RC3) · Anonymous Referee #3 · 7 Oct 2020

Review of "Effect of NOX, O3 and NH3 on sulfur isotope composition during heterogeneous oxidation of SO2: a laboratory investigation" by Guo et al.

The topic of the paper is relevant for interpretation of isotope ratios of sulfate in the atmosphere. Unfortunately, the presentation and analysis of the paper is insufficient in my view for publication in a scientific journal. This relates to many categories that I consider important for a scientific publication, and I can only list a few points here:

Maybe most importantly, for all key figures, the numbers in the text disagree with what I see in the figures.

Examples:

L 104: As shown in Fig. 2, the yield of SO42- ranged from 0.0097 to 0.7795 g and the values of  $\delta 34$  S were 2.9–4.8 %

I see no sulfate concentration above 0.6 g in figure 2, and no delta values below 3.5 %

L129/130: The yield of SO42- ranged from 0.0081 to 0.6712 g with the  $\delta$ 34S values of 1.6–2.9 ‰ (Fig. 3).

I see no sulfate concentration above 0.6 g in figure 3

Line 157: yield of SO42- ranged from 0.0237 to 0.9469 g with the  $\delta 34$  S values of 0.8–4.3 ‰ (Fig. 4).

I neither see the very small or very high sulfate values mentioned in the text in Fig. 4. The range in figure 4 (1.2-2.5 % is very different from what is written in the text.

I wonder what is going on there. This looks like either the authors have not checked the consistency between text and figures and need reviewers to find this out, which is not acceptable, or a crucial piece of information is missing.

The methods section provides almost no information on experimental conditions and setup, I have to guess how experiments were conducted. I wonder whether there is any relation of the samples collected outside and the lab experiments. I understand from the description that the concentrations of the reactants are in the % range, is this then representative for the atmosphere? Why were the various flow rates used? How are samples collected in the lab experiments? These are only a few questions.

In the discussion, the authors completely mix the interpretation of their results with what was found in previous studies and it is not possible to clearly understand when they are reporting a result from a previous study and when they put the results of their experiments in light of previous findings. I think this paper would require a clear separation of results and discussion. And then still a careful distinction on what is new from the present paper and what has been found before.

Quantitatively, the epsilon value of the overall process (1.3 % line 192), which enables source partitioning, falls completely from heaven. Where does it come from? Does the evaluation include the assumption from the literature that the source isotopic composition is 2.7 % +- 2 % (line 184)? That would result in a huge error. In the present version of the paper, Figure 5 shows some synthetic Rayleigh fractionation curves, but I see no relation to the data, other than the unexplained value of 1.3 %

All the data shown in Figure 1-4 should have associated error bars.

СЗ

---

## Author Comment (AC1)

Thank you very much for your kind consideration and for the comments concerning our manuscript entitled "Effect of $NO_X$, $O_3$ and $NH_3$ on sulfur isotope composition during heterogeneous oxidation of $SO_2$: a laboratory investigation". Those comments are all valuable and helpful for improving our paper, as well as providing important guiding significance to our future research.

We have considered these comments carefully and we have made some modifications. We narrowed the focus and only focused on the effects of $NO_X$, $NH_3$, and $O_3$ on $SO_2$ oxidation on the surface of mineral dust. So we only calculated the relative contribution of $NO_X$, $NH_3$, and $O_3$ to the oxidation of $SO_2$ on mineral dust, not taking the contribution of TMI oxidation into account. Besides, we calculated the relative contribution on the basis of sulfur isotope fractionation factors rather than the sulfur isotope values. The sulfur isotope compositions were investigated to determine the effect of $NO_X$, $O_3$ and $NH_3$ on the heterogeneous oxidation of $SO_2$ on $\alpha$-$Fe_2O_3$. Moreover, the data of the actual observation was only used for comparison with results in experiments. The observational data showed that there existed fractionation effects in the process of $SO_2$ oxidation in the atmosphere. Hence, sulfur isotope fractionation can be used to investigate the effects of $NO_X$, $O_3$, and $NH_3$ on the heterogeneous oxidation of $SO_2$ on the $\alpha$-$Fe_2O_3$ surface in the lab experiments. And we solved the equation by the linear fitting. Additionally, hematite is the main component that can oxidize $SO_2$ in atmospheric real mineral. Hence, we chose hematite as the main mineral in our experiments. Little $Fe^{3+}$ can be dissolved from hematite in the presence of aerosol water in our experiments. So the aqueous-phase oxidation may occur on the surface of hematite. We also found some grammatical issues and have revised them in the revised manuscript.

Revised portions are marked in red in the revised paper. The responses to Reviewers' comments as well as the main corrections in the revised paper are as follows.

**Response to Reviewer#1's comments**

-I think this study adds information on the sulfur isotopic fractionation ratio ($\delta^{34}S$) from several heterogeneous $SO_2$ oxidation pathways involving hematite with $NO_X$, ozone, and $NH_3$. However, I do not think that the results presented here allow the authors to draw conclusions about the relative importance of different pathways in real field observations. This is my largest concern. The authors only include the three heterogeneous pathways they measured in this study's laboratory experiments when interpreting observed $\delta^{34}S$ using the Rayleigh distillation model. However, there are also other aqueous-phase and gas-phase reactions with different fractionation factors (e.g. Harris et al., 2012a, b; who also show it is not possible to distinguish between some pathways based on S34 alone). For example, Shao et al. (2019) found heterogeneous oxidation of $SO_2$ on all aerosols was only 20% in wintertime in China, with alkaline dust (for which hematite is possibly an analog) contributing a total of only 9%. Even without considering additional reactions which would create a very underconstrained problem with no solution, the authors are solving for three unknowns (the relative importance of their three studied reactions) using only two equations (equation 3 and line 199). How is this possible? I am a bit confused that the authors do not show or use the concentrations of potential oxidants measured at the field site to address the relative importance of different pathways, although they are vaguely referred to in lines 202-203.

**Response:** Thank Reviewer#1 a lot for suggestions. Indeed, there are also other aqueous-phase and gas-phase reactions with different fractionation factors. As a result, we narrowed the focus to the

heterogeneous oxidation of $SO_2$. The efficient conversion of $SO_2$ to $SO_4^{2-}$ occurs at high RH in the presence of a high concentration of $NO_X$ and $NH_3$ (Wang et al., 2016). And the high concentration of $O_3$ derived from the atmospheric photochemistry due to decreases in the concentration of $PM_{2.5}$ (Cheng et al., 2016). According to these literatures, $NO_X$, $NH_3$, and $O_3$ have non-negligible effects on the oxidation of $SO_2$. However, these studies did not take into account the influences of mineral surfaces on $SO_2$ oxidation. Hence, we only focused on the impacts of these three gases on $SO_2$ oxidation on the surface of mineral dust and calculated their relative contributions. Besides, we calculated the relative contribution based on sulfur isotope fractionation factors for distinguishing the specific oxidation path of $SO_2$. Finally, the data of the actual observation vaguely referred to was only used for comparison with results in experiments. The observational data showed that there existed fractionation effects in the process of $SO_2$ oxidation in the atmosphere. Hence, sulfur isotope fractionation can be used to investigate the effects of $NO_X$, $O_3$, and $NH_3$ on the heterogeneous oxidation of $SO_2$ on $\alpha$-$Fe_2O_3$ surface in the lab experiments. We solved the equation by the linear fitting. Also, we found some grammatical issues and have revised them as following for more clarity.

**Page 2, Lines 60:**

Nevertheless, $\delta^{34}S$ values can only be used to roughly distinguish between the heterogeneous and homogeneous oxidation. The specific oxidation path of $SO_2$ cannot be separated due to the similarity of the sulfur isotope values under different conditions during the oxidation processes of $SO_2$ (Harris et al., 2012b). Hence, the distinctive isotope fractionations for different oxidation processes of $SO_2$ can be applied in the study of the formation processes of sulfate.

**Page 2, Lines 47:**

$SO_2$, $NO_X$, and $NH_3$ synergistically accelerated the formation of sulfate under high humidity (Wang et al., 2016). The oxidation by $NO_2$ is more dominant than that by $O_3$ at high pH values (Cheng et al., 2016).

**Page 4, Lines 109:**

The variable $\delta^{34}S_{aerosol}$ values were attributed to the fractionation effects during the process of $SO_2$ oxidation (Li et al., 2020). Hence, sulfur isotope fractionation can be used to investigate the effects of $NO_X$, $O_3$, and $NH_3$ on the heterogeneous oxidation of $SO_2$ on $\alpha$-$Fe_2O_3$ surface in the lab experiments.

**Page 7, Lines 221:**

The contributions of $NO_X$, $O_3$, and $NH_3$ pathways determined by Eq. 2 and linear regression were 67.5±10 %, 13.3±10 %, 19.2±10 %, respectively.

-I think the authors should also explain in more detail why hematite is chosen. Is it a true analog of mineral dust? How do the reactions involving hematite in this paper relate to the aqueous-phase transition metal ion catalyzed $SO_2$ oxidation discussed at length in the literature? Is there aerosol water present at RH=40% and thus the possibility of aqueous-phase oxidation reactions in hematite leachate? How does this study of fractionation in reactions with only iron-containing hematite relate to/inform interpretation of previous results for actual desert dust (e.g., Harris et al., 2012b)? I think these questions should be addressed explicitly in the paper.

**Response:** Thank for Reviewer#1 thoughtful suggestions. Hematite is not a true analog of mineral dust. There are many minerals in a true analog of mineral dust. However, iron is one of the most important components in the mineral aerosol. The major source of iron in the aerosol is in the form of highly insoluble iron oxides (Fu et al., 2007; Usher et al., 2003). And hematite is the main representative of mineral that can oxidize $SO_2$. So we chose hematite in our experiments. In addition, little $Fe^{3+}$ can be dissolved from hematite due to the presence of aerosol water in our experiments. Hence, the reactions in our study may include the catalytic oxidation. Besides, there existed aerosol water when RH was 40% (Xu et al., 2015). So aqueous-phase oxidation reactions may occur on the surface of hematite in the presence of aerosol water. Moreover, we further confirmed the experimental conclusion of Harris et al. (2012b). For more clarity, the following statements have been included in the revised manuscript:

**Page 3, Lines 53:**

Mineral dust is a major fraction of global atmospheric aerosol, and the oxidation of $SO_2$ on mineral dust has implications for cloud formation, climate and the sulfur cycle (Harris et al., 2012a). Understanding the oxidation of $SO_2$ on mineral dust is a key part of investigating the interactions between dust, sulfur, and clouds.

**Page 3, Lines 70:**

Besides, as one of the typical mineral oxides, hematite) can accelerate the conversion rate of $SO_2$ to sulfate (Usher et al., 2003; Fu et al., 2007).

**Page 3, Lines 94:**

Aqueous-phase oxidation reactions may occur on the mineral surface due to the presence of aerosol water (Xu et al., 2015).

Three new references have been cited in the revised paper.
**Reference:**
Fu, H., Wang, X., Wu, H., Yin, Y., Chen J.: Heterogeneous Uptake and Oxidation of SO2 on Iron Oxides, J. Phys. Chem. C., 111, 6077–6085, doi:10.1021/jp070087b, 2007.
Usher, C. R., Michel, A. E., and Grassian, V. H.: Reactions on Mineral Dust, Chem. Rev., 103 (12), 4883–4940, doi:10.1021/cr020657y, 2003.
Xu, W., Li, Q., Shang, J., Liu, J., Feng, X., Zhu, T.: Heterogeneous oxidation of SO2 by O3-aged black carbon and its dithiothreitol oxidative potential, J. Environ. Sci., 36, 56–62, doi:10.1016/j.jes.2015.02.014, 2015.

- Line 37: "the formation mechanism of sulfate is still unclear": is this globally, during pollution/extreme haze events, or in China specifically? Given that there is no discussion of, e.g., DMS oxidation, or oxidations by HOCl/HOBr (Q. Chen et al. (2017), GRL), I think the focus in the introduction should be narrowed appropriately and it would make more clear why which studies are cited in this introduction.

**Response:** We completely agree with this valuable suggestion. It is the formation mechanism of sulfate during haze events in China that is still unclear. The studies about the influence of $NO_X$, $O_3$, and $NH_3$ on $SO_2$ oxidation were cited in our manuscript, which was to prove that these gases play an important

role in the oxidation of $SO_2$. However, these studies did not take into account the impacts of mineral surfaces on $SO_2$ oxidation. So we narrowed the focus and only focused on the effects of $NO_X$, $O_3$, and $NH_3$ on the formation mechanism of sulfate on mineral dust. The relevant descriptions have been included in the revised manuscript.

**Page 2, Lines 44:**

Therefore gaseous oxides are the key oxidation pathways in the formation of sulfate during the heterogeneous oxidation process (Chen et al., 2017a).

$NO_X$, $O_3$, and $NH_3$ have been confirmed that they have non-negligible effects and contributions to the formation mechanism of sulfate. $SO_2$, $NO_X$, and $NH_3$ synergistically accelerated the formation of sulfate under high humidity and the oxidation by $NO_2$ oxidation is more dominant than that by $O_3$ at high pH values (Wang et al., 2016; Cheng et al., 2016). Li et al. (2018b) demonstrated that the oxidation of S(IV) in haze in China was driven by the $HONO/NO_2^-$ generated due to the consumption of $NO_2$ on the surface of aerosols. Besides, aqueous $NO_2$ serves as the dominant oxidant of $SO_2$ at highly elevated $NO_X$ levels (Xue et al., 2019). The synergistic effect between $NO_2$ and $SO_2$ on the surface of mineral promotes the conversion of $SO_2$ to sulfate (Ma et al., 2018). However, these studies did not take the influence of mineral dust on $SO_2$ oxidation into consideration. Mineral dust is a major fraction of global atmospheric aerosol, and the oxidation of $SO_2$ on mineral dust has implications for cloud formation, climate and the sulfur cycle (Harris et al., 2012a). Understanding the oxidation of $SO_2$ on mineral dust is a key part of investigating the interactions between dust, sulfur, and clouds.

**Page 3, Lines 71:**

Herein, for the first time, the several $SO_2$ oxidation processes with the different chemical conditions ($NO_X$, $O_3$, and $NH_3$) are conducted on the hematite surface in the laboratory to gain insight into the sulfur isotope fractionation.

- Line 39: Li et al. (2018): there are two citations that are Li et al. (2018), please specify which one.

**Response:** Sorry for this mistake. We have revised this sentence as follows.

**Page 2, Lines 48:**

Li et al. (2018b) demonstrated that the oxidation of S(IV) in haze in China was driven by the $HONO/NO_2^-$ generated due to the consumption of $NO_2$ on the surface of aerosols.

- Line 42: please add a citation to this sentence ("the synergistic effect. . . promotes the conversion of $SO_2$ to sulfate") or make more clear if it is related to previous citations in this paragraph.

**Response:** Thank you for the suggestion. The citation was added as follows.

**Page 2, Lines 51:**

The synergistic effect between $NO_2$ and $SO_2$ on the surface of mineral promotes the conversion of $SO_2$ to sulfate (Ma et al., 2018).

- Line 45-46: "measured sulfur isotopic fractionation"- could you please specify which isotopic ratio

this paper is studying (e.g., is it $\delta^{34}S$ or something else)?

**Response:** We are very sorry for our cursoriness. Here should be sulfur isotopic composition ($\delta^{34}S$) rather than sulfur isotopic fractionation.

**Page 2, Lines 39:**

The measured sulfur isotopic composition ($\delta^{34}S$) showed that about -9 ‰ is for homogeneous oxidation of $SO_2$ and up to +16.5 ‰ is for heterogeneous oxidation of $SO_2$ (Chen et al., 2017a).

- Line 51: there are two Chen et al. (2017) papers in the references, please specify which one.

**Response:** Thank you. The citations were specified as follows.

**Page 2, Lines 39:**

The measured sulfur isotopic composition ($\delta^{34}S$) showed that about -9 ‰ is for homogeneous oxidation of $SO_2$ and up to +16.5 ‰ is for heterogeneous oxidation of $SO_2$ (Chen et al., 2017a).

- Line 59-60: please add a citation to this sentence (e.g., Harris et al., 2012b)

**Response:** Thank you for your suggestion. We added the citation (Harris et al., 2012b) in the revised manuscript.

**Page 2, Lines 61:**

The specific oxidation path of $SO_2$ cannot be discriminated due to the similarity of the sulfur isotope values under different conditions during the oxidation processes of $SO_2$ (Harris et al., 2012b).

- Line 70: I would reword "the causes of subsequent sulfates" to something like "the subsequent formation of sulfate"

**Response:** We are very sorry for the ambiguity. We carefully check this sentence and corrected the grammar.

**Page 3, Lines 74:**

It may not only provide a theoretical basis for the subsequent formation of sulfate but also be crucial for improving the air quality and studying the regional climate change.

- Line 83-84: could you state the average temperature and RH at the sample site? I don't know what "similarly" means in this context.

**Response:** Thank you for the suggestion. The average temperature (293K) and RH (60%) at the sample site were added in this paragraph.

**Page 3, Lines 95:**

The values of temperature and humidity in the experiments were set similarly to those in the air

(293 K and 60 %) during this period.

- Section 2.1: add a sentence or two to very clearly distinguish between the field sampling vs. the laboratory experiments, e.g., a beginning sentence for each of the first two paragraphs of this section. In fact, I would move the sentence in lines 95-96 in Section 3 to the first paragraph, and the sentence in line 100-102 to the 2nd paragraph.

**Response:** Thank you for the valuable suggestions. Sentences to distinguish between the field sampling and the laboratory experiments were added. We adopted and implemented your suggestion to move the sentence in lines 95-96 in Section 3 to the first paragraph, and the sentence in line 100-102 to the 2nd paragraph as following.

**Page 3, Lines 79:**

First all, for confirming the sulfur isotope composition and fractionation, $PM_{2.5}$ samples from $26^{th}$ Feb. to $6^{th}$ Apr. 2016 were collected and the $\delta^{34}S$ values of them were measured. The sampling site was located on the roof of the library in Nanjing University of Information Science & Technology (32.1°N, 118.5°E). $PM_{2.5}$ samples were collected by a high volume sampler (TH-1000H, Tianhong Co., Wuhan) with a flow rate of 1.05 $m^3$ $min^{-1}$ from 9 am to 9 pm per day. The observational data were only used for comparison with results in experiments.

To shed light on the mechanism of $SO_2$ oxidation in sulfur isotope fractionation and sulfate formation, the $SO_2$ oxidation processes on the surface of $\alpha$-$Fe_2O_3$ in the presence of $NO_X$, $O_3$, and $NH_3$ were carried out in the laboratory.

Figures:
- All figures: it would be very helpful if Figure 2a,3a,4a and 2b,3b,4b had the same y-axis range. It makes it harder to compare across oxidants otherwise. Is each point based on a single measurement, or are multiple samples taken? If the latter, the standard deviation/error should be shown. If not, I think it would be helpful to add either perpendicular lines on the points or shading to indicate the accuracy of the measurements, as ±0.2 per mil is as large as some of the changes in $\delta^{34}S$ caused by changing the volume ratios of oxidants. I believe the y-axis label should be changed from to $\delta^{34}S$.

**Response:** Thank for your suggestions. We have made the same y-axis range for Figure 2a,3a,4a and 2b,3b,4b and added perpendicular lines to indicate the accuracy of the measurements. Simultaneously, the y-axis label was changed to $\delta^{34}S$.

- Figures 2 and 3: what kind of curve-fitting is being done here? Splines? A polynomial function? I think the nonlinear curves mislead the reader to think that there is a change in the isotopic fractionation between measured values that we cannot know for sure from the experiments performed here alone (e.g., the "both $O_2$+light" blue line in figure 2b between 1:1 and 1:4 ratio makes it seem like it is decreasing, but that is a product of the nonlinear curve-fitting being done with only 3 points. maybe it's actually flat or slightly increasing if it was a linear trend instead). I would either remove the lines entirely and only show points, or simply draw lines between each set of points as this is the simplest type of fitting to do and there is such limited data to justify a polynomial fit.

**Response:** We are very sorry for the ambiguity. The curve-fitting was splines. We changed the splines to simply draw lines between each set of points.

\- Figure 3 caption: "(The dotted line represents the trend as it should be)." What does this mean? This is not explained in the text. I think the dotted line should be removed unless it can be explained why this would be known from theory.

**Response:** Thank you. The dotted line was the results we previously expected. We have measured the data at 1:1 ratio and added to the figure. We took your meaningful suggestion that the dotted line was removed.

\- Figure 4: I would make this a line graph instead of a bar chart, I am not sure why it is different from Figures 2 and 3. Also, I would remove the arrows from Figure 4a, I don't know what this adds. Explain why you did not do a 1:1 ratio experiment (e.g., it requires ammonia in excess).

**Response:** We are very sorry for our cursoriness. We made this a line graph instead of a bar chart and removed the arrows in Figure 4. The $\delta^{34}S$ values of sulfate increased with the increases of sulfate concentrations in the presence of $O_2$ and light according to Doi et al. (2004). We wrongly think that there were synergistic effects between $O_2$ and light. So we followed your suggestions and carried out the supplementary experiment and added the data of 1:1 ratio in the revised manuscript. After adding the data, the overall trend of our results showed a counteracting effect between $O_2$ and light.

**Page 13-14, Lines 431-436:**

[Figure]

**Figure 2:** Effect of the mixed gas of $SO_2$ and $NO_X$ on (a) sulfate production and (b) sulfur isotope value under different reaction conditions. The perpendicular lines on the points indicate the accuracy of the measurements.

[Figure]

**Figure 3:** Effect of the mixed gas of $SO_2$ and $O_3$ on (a) sulfate production and (b) sulfur isotope values under different reaction conditions. The perpendicular lines on the points indicate the accuracy of the measurements.

[Figure]

**Figure 4:** Effect of the mixed gas of $SO_2$ and $NH_3$ on (a) sulfate production and (b) sulfur isotope values under different reaction conditions.

$NO_X$ results description:

- Lines 103-104 (First sentence of this paragraph): To me, the way this is currently worded makes it unclear whether this is something known before or based specifically on the results of this study and what is shown in Figures 2 to 4. What do you mean by taking ion strength into account? How is that accounted for in the presentation of Figures 2-4?

**Response:** Thank you for the suggestion. "taking ion strength into account" was based on the result of Cheng et al. (2016). So we added the citation of Cheng et al. (2016) for $NO_X$. "taking ion strength into account" was to prove that $NO_X$ is the most important oxidant during the heterogeneous oxidation of $SO_2$. As we did not calculate the ion strength in our experiments, this cannot be accounted for in Figures 2-4. For more clarity, we have revised the sentence as follows.

**Page 4, Lines 113:**

$NO_X$ has been determined that it is the most important oxidant during the heterogeneous oxidation of $SO_2$ taking the impact of ion strength of $SO_4^{2-}$ into account by Cheng et al. (2016).

- Lines 109-110: "In addition, the increase in the amount of $NO_X$ was another key factor that led to the acceleration of sulfate formation (Cheng et al., 2016)." It is not clear whether this refers to results from

your study, Cheng et al. (2016), or both. Please revise to clarify.

**Response:** We are very sorry for the ambiguity. The results of our study and Cheng et al. (2016) both illustrated this view. We revised the sentence as follows.

**Page 4, Lines 119:**
In addition, the results in our study supported this view found by Cheng et al. (2016) that the increase in the amount of $NO_X$ was another key factor that led to the acceleration of sulfate formation.

- Lines 123-124: "Therefore, under dark conditions, the sulfur isotope value was mainly affected by oxygen and the catalytic action of Fe(III), resulting in the enrichment of lighter sulfur isotopes (Han et al., 2016a)." Similar to the above comments. Please clarify if this is the case for both your experiments and previous work; if not, please state that explicitly.

**Response:** Thank you for the suggestion. The sentence was the result of our experiments. Thus we deleted the citation to make it more clear.

**Page 4, Lines 134:**
Therefore, under dark conditions, the sulfur isotope value was mainly affected by oxygen and the catalytic action of Fe(III), resulting in the enrichment of lighter sulfur isotopes.

- Line 149-151: I think Harris et al. (2012a) should be cited here, which is about OH in the gas-phase. I am not sure though how OH in the gas phase would relate to heterogeneous oxidation here. If I am mistaken, more explanation is needed of how Harris et al. (2012b) relates to your study and OH.

**Response:** Thank for your suggestion. •OH in our experiments was produced by photolysis of ozone. The citation of Harris et al. (2012b) was used to support that •OH can promote the enrichment of heavy sulfur isotopes. The sentence was revised as follows.

**Page 5, Lines 159:**
The high $\delta^{34}S$ values, especially the highest value obtained at the ratio of 1:8, may be in relation to •OH produced by photolysis of ozone (Wang et al., 2019). •OH has been demonstrated by Harris et al. (2012b) that can promote the enrichment of heavy sulfur isotopes.

- Line 154: specify which Harris et al. 2012 paper.

**Response:** We are very sorry for this ambiguity. We revised the manuscript and changed Harris et al. 2012 to Harris et al. 2012b.

**Page 5, Lines 165:**
Harris et al. (2012b) confirmed the hypothesis that equilibration to higher pH increased fractionation.

Ozone results description:

- Reactions R4 & R5: the sentence before says, "which was described as". Please give the citation where these reactions were described.

**Response:** Thank you. The citation (Cheng et al., 2016) was added to the sentence.

**Page 5, Lines 140:**
   Ozone, as a very efficient oxidant, could react with the sulfite to release oxygen, promoting the subsequent oxidation of $SO_2$ (Cheng et al., 2016):

$SO_3^{2-} + O_3 \rightarrow SO_4^{2-} + O_2,$                                     (R4)

$HSO_3^- + O_3 \rightarrow HSO_4^- + O_2,$                                     (R5)

- Line 136-137: what mineral did Nicoals et al. (2019) use? Would it behave the same as the hematite used in your study?

**Response:** Thank you for suggestions. $TiO_2$ was used by Nicoals et al. (2019). Hematite and $TiO_2$ behave the same that both can chemisorb $SO_2$ to a bidentate complex (Harris et al., 2012a).

- Lines 138-139: "Under dark conditions, photolysis of $O_3$ were negligible, thus surface reactions will be solely responsible for sulfate production (Harris et al., 2013a)." I find this sentence confusing. I think I understand what you mean, but please restate so it is clear that during your study and in general, there is no photolysis of ozone in the dark, so any sulfate production during the dark does not involve the ozone photolysis pathway. I think it would make more sense to introduce the ozone photolysis pathway (lines 140-147) first and then interpret your experimental results in the light and dark.

**Response:** We completely agree with this valuable suggestion. We revised this paragraph to move the ozone photolysis pathway to the front of the experimental results.

**Page 5, Lines 148:**
   Moreover, the photolysis of ozone under UV radiation formed electronically excited $O(^1D)$, and its subsequent reaction with water vapor could generate a mass of hydroxyl radicals (Ran et al., 2014; Cheng et al., 2016). As adsorption sites for water, surface hydroxyls were the principal reactive sites on metal oxides. In turn, the adsorbed water was either dissociated into more hydroxyls at oxygen vacancies or hydrogen-bonded to surface O-H groups, which was in favor of the heterogeneous oxidation of $SO_2$ (Wang et al., 2019).

$O_3 + h\upsilon \rightarrow O(^1D) + O_2,$                                     (R6)

$O(^1D) + H_2O \rightarrow 2\bullet OH,$                                     (R7)

$\bullet OH + SO_2 + M \rightarrow HO_2 + SO_4^{2-},$                                     (R8)

   Hence, under dark conditions, photolysis of $O_3$ was negligible, thus oxidation by $O_3$ will be responsible for sulfate production (Harris et al., 2013a)

- Line 153-154: "In addition, uptake and decomposition of ozone under irradiation increased the basicity of the surface, which was conducive to enrich heavy sulfur isotopes (Hanisch et al., 2003)." Again, reword to clarify what you are assuming to apply to your study vs. what you actually can determine from your results. As an example, something like this would help clarify for me as a reader:

"It is possible that the uptake and decomposition of ozone under irradiation in our study would increase the basicity of the surface, which Hanisch et al. (2003) found to increase enrichment of heavy sulfur isotopes and is consistent with the increased enrichment in our irradiated ozone experiments."

**Response:** We are very sorry for the ambiguity. We took your suggestion and changed the sentence to "It is possible that the uptake and decomposition of ozone under irradiation in our study would increase the basicity of the surface, which Hanisch et al. (2003) found to increase enrichment of heavy sulfur isotopes and is consistent with the increased enrichment in our irradiated ozone experiments." The following statements have been included in the revised manuscript:

**Page 5, Lines 163:**
It is possible that the uptake and decomposition of ozone under irradiation in our study would increase the basicity of the surface, which Hanisch et al. (2003) found to increase enrichment of heavy sulfur isotopes and is consistent with the increased enrichment in our irradiated ozone experiments.

$NH_3$ results description (lines 156 - 175)
- Line 158-160: "The extent of aerosol neutralization was determined principally by the ambient concentration of $NH_3$ and the oxidation rate of $SO_2$ (Kong et al., 2019)." I don't know if this sentence is needed. In your study, you are not determining the aerosol neutralization, correct? To me, it sounds like it is referencing the methods of the previous study (Donaldson et al., 2010), which described the neuralization of the ammonium salts. I am not sure if this is what you meant.

**Response:** We are very sorry for our thoughtlessness. This sentence was the result found by Donaldson et al. (2010) and we didn't determine the aerosol neutralization. As a result, the sentence was deleted in the revised manuscript.

- Line 163-164: "Moreover, the oxygen vacancies in $\alpha$-$Fe_2O_3$ may lead to the formation of sulfate on $\alpha$-$Fe_2O_3$ (Wang et al., 2019)." Is this only for the ammonia experiments specifically, or would this statement actually apply to all of your laboratory experiments? Please explain.

**Response:** Thank for your reasonable comments. The sentence is only for the ammonia experiments and not suitable for all of our laboratory experiments. The sentence was revised as follows.

**Page 6, Lines 176:**
The oxygen vacancies in $\alpha$-$Fe_2O_3$ may lead to the formation of sulfate on $\alpha$-$Fe_2O_3$ in the presence of $NH_3$ (Yang et al., 2016).

The new reference has been cited in the revised paper.
**Reference:**
Yang, W., He, H., Ma, Q., Ma, J., Liu, Y., Liu, P., and Mu, Y.: Synergistic formation of sulfate and ammonium resulting from reaction between $SO_2$ and $NH_3$ on typical mineral dust, Phys. Chem. Chem. Phys., 18 (2), 956–964. doi:10.1039/c5cp06144j, 2016.

- line 166-167 "Under only-light, $NH_3$, which increased the alkalinity by producing OH from

hydrolysis, dominated in the reaction, leading to an increase of $\delta^{34}S$ values (Jiang et al., 2017)." : Please separate out which part of this sentence is based on the results found here vs. the part that is coming from Jiang et al. (2017). My guess would be that the $\delta^{34}S$ part is the results from this study, and the alkalinity by producing OH from hydrolysis is from Jiang et al. (2017), but it is not obvious to me as a reader.

**Response:** Thank you. The increase of $\delta^{34}S$ is the result of this study, and the alkalinity by producing OH from hydrolysis is from Jiang et al. (2017). We revised the sentence to make it more clarified.

**Page 6, Lines 184:**

    Owing to the increase of alkalinity by producing $OH^-$ from hydrolysis of $NH_3$ (Jiang et al., 2017), $NH_3$ dominated in the reaction under only-light, leading to an increase of $\delta^{34}S$ values in our study.

- Line 167-168: "$O_2$ with $Fe^{3+}$ as catalyst dominant in the presence of only-oxygen favored light sulfur isotopes, which was consistent with above results." Are the "above results" from Jiang et al. (2017), or the $\delta^{34}S$ results found in your laboratory experiment? Can you please explain more clearly why $\delta^{34}S$ decreases during only-$O_2$ experiments when ammonia is increased but not when ozone or $NO_X$ are increased? How does ammonia affect the transition-metal-ion catalyzed pathway?

**Response:** Thanks a lot for your suggestions. The "above results" were the $\delta^{34}S$ results found in our laboratory experiment and we revised the sentence as follows. Besides, many factors lead to the decrease of the $\delta^{34}S$ values in the presence of $NH_3$ rather than $NO_X$ or $O_3$, such as the different redox properties and pressure. So we can't clearly point out what causes the decrease of $\delta^{34}S$. Moreover, it is still unclear that how ammonia affects the transition-metal-ion catalyzed pathway. However, this will provide a direction for our further research.

**Page 6, Lines 185:**

    Similar to the sulfur isotope values in the presence of $NO_X$ and $O_3$, $O_2$ with $Fe^{3+}$ as catalyst dominant under only-oxygen favored light sulfur isotopes in the presence of $NH_3$.

- Line 168-169: "The $\delta^{34}S$ values of sulfate increased with the increases of sulfate concentrations under the combined oxygen with light (Doi et al., 2004)." Is this the results of Doi et al. (2004) for ammonia experiments? This seems to conflict with your results, which show in Fig. 4b that $\delta^{34}S$ stays the same under the combined oxygen with light for higher ammonia (which caused higher sulfate production in figure 4a).

**Response:** We are so sorry for the thoughtlessness. The results of Doi et al. (2004) were for ammonia experiments. Besides, we followed your suggestions that we did the supplementary experiment and added the data of 1:1 ratio in the revised manuscript. We found that the $\delta^{34}S$ values of sulfate slightly increased with the increases of sulfate concentrations, which was consistent with the results of Doi et al. (2004). So we revised the sentence as follows.

**Page 5, Lines 187:**

    The $\delta^{34}S$ values of sulfate slightly increased with the increases of sulfate concentrations under the

combined oxygen with light.

- Line 169-170: "Therefore, we inferred that $O_2$ and light had a synergistic effect on the sulfur isotope compositions in the presence of $NH_3$." I am not sure how this relates to the previous sentence citing Doi et al. (2004). To me it seems that the effects cancel each other out: under increasing ammonia, the "only $O_2$" experiment has decreased $\delta^{34}S$ while the "only light" experiment has increased $\delta^{34}S$ and the "both $O_2$ + light" experiment has nearly constant $\delta^{34}S$. Perhaps a different word besides "synergistic" would be more specific.

**Response:** Thank you for the suggestions. The effects of the only $O_2$ and only light indeed cancel each other out to a certain extent, so that the $\delta^{34}S$ values were nearly constant under both $O_2$ and light. So we changed the word "synergistic". For more clarity, the following statements have been included in the revised manuscript:

**Page 6, Lines 188:**
  Therefore, we inferred that the impacts of $O_2$ and light on the sulfur isotope compositions offset each other to some extent in the presence of $NH_3$.

- Line 171-173: "The $\delta^{34}S$ values from main biogenic source of $NH_3$ were on the low side, indicating that the effects of $NH_3$ on sulfur isotopic compositions were relatively mild (Han et al., 2016a; Grewling et al., 2019)." Again, I do not know what part of this sentence is based on the results of your laboratory experiments here vs. these previous studies that are cited. What does the biogenic nature of $NH_3$ have to do with your experiments?

**Response:** We are very sorry for this ambiguity. The sentence was all based on the previous studies that are cited. "the effects of $NH_3$ on sulfur isotopic compositions were relatively mild" was based on the results of our experiments. Besides, the biogenic nature of $NH_3$ has nothing to do with our experiments. So we deleted "biogenic source of $NH_3$". The relevant descriptions have been included in the revised manuscript.

**Page 6, Lines 190:**
  The effects of $NH_3$ on sulfur isotopic compositions were also proved to be relatively mild by Grewling et al. (2019).

Rayleigh distillation and enrichment, lines 176-205
- Line 181 (equation 1): It is not clear to me if this equation is needed as it seems it is not used and instead equation 2 (line 289) is. If it is used, can it be made more clear how and why the two equations are presented?

**Response:** Thanks a lot for your suggestion. The two equations were put in the manuscript previously because we want to make a comparison to highlight the modified equation 2. We took the suggestion that equation 1 was deleted after careful consideration of Reviewer#1 suggestion.

- Line 193 and Figure 5: what does "simulations" mean exactly? What data is specifically being plotted

here? What do the shaded regions around each line mean? Please add that to the figure caption. A standard deviation is given for the emissions, but not the other lines.

**Response:** Thank you for the reasonable comments. "simulation" means measurements in laboratory experiments. The experiments of mixing three gases ($NO_X$, $O_3$, and $NH_3$) were also conducted in the laboratory. The epsilon value of 1.3 ‰ was the average value in this experiment. The measured $\delta^{34}S_{obs}$ and $\delta^{34}S_{sulfate}$ values under different oxidation paths in this study were plotted here. The shaded regions around each line mean the measurement error. The explanation was added to the figure caption. We also added some standard deviations for $\delta^{34}S$ values. For more clarity, the following statements have been included in the revised manuscript:

**Page 7, Lines 211:**

The $\varepsilon_{obs}$ values as a result of mixing oxidation pathways of $SO_2$ + $NO_X$ + $O_3$ + $NH_3$ were +1.3±1.4 ‰ measured in our laboratory experiments of mixing three gases ($NO_X$, $O_3$, and $NH_3$) (Fig. 6).

**Page 15, Lines 437-440:**

[Figure]

**Figure 5:** Rayleigh distillation model of sulfate production. The red circles are the measured $\delta^{34}S_{obs}$ in this study. Dashed lines with shaded areas are $\delta^{34}S_{sulfate}$ values: the red line indicates the $\delta^{34}S_{sulfate}$ (+4.5±0.5 ‰) when $SO_2$ is oxidized solely by $NO_X$, the blue line indicates the $\delta^{34}S_{sulfate}$ (+2.2±0.3 ‰) when $SO_2$ is oxidized solely by $O_3$, the green line indicates the $\delta^{34}S_{sulfate}$ (+1.6±1.3 ‰) when $SO_2$ is oxidized solely by $NH_3$, and the dark grey line is the $\varepsilon_{obs}$ of +1.3±1.4 ‰. The shaded regions around each line mean the measurement error

- Equation 3, Lines 199-202: Please explain how three unknowns are solved with only two equations.

**Response:** Thank you for the suggestion. We used the linear regression method for fitting to solve the equations.

**Page 7, Lines 221:**

The contributions of $NO_X$, $O_3$, and $NH_3$ pathways determined by Eq. 2 and linear regression were 67.5±10 %, 13.3±10 %, 19.2±10 %, respectively.

**Response to Reviewer#2' s comments**

- I have read this manuscript, and I found that I mostly agree with previous comments from referee #1. This study provides sulfur isotopic fractionation for sulfate formation from $SO_2$ in the presence of $NO_X$, $O_3$, and $NH_3$. Although these experimental results show some interesting phenomena, I do not think that these results lead to the conclusions drawn by the authors. Note that $SO_2$ has two oxygen atoms and $SO_4^{2-}$ has four oxygen atoms, thus we have to think the origins of oxygen atoms in sulfate formation. The effect of $NH_3$ for sulfate formation is interesting, because the presence of $NH_3$ may change pH in liquid and promote the pH-dependent process such as $O_3$, TMI, $NO_2$. Unfortunately, I found a lack of this viewpoint throughout this manuscript.

**Response:** We completely agree with the valuable suggestions. Indeed, in the presence of $NH_3$, $NH_3$ was not an oxidant to convert $SO_2$ to $SO_4^{2-}$. However, there was dissolved oxygen in the presence of aerosol water (Chu et al., 2016), so the process consisting of oxidation of $HSO_3^-$ or $SO_3^{2-}$ to $SO_4^{2-}$ may be involved in the formation of sulfate. Baltrusaitis et al. (2007) also proposed a mechanism for the oxidation of $SO_2$ to sulfate involving the reaction of adsorbed sulfite with activated molecular oxygen at $\alpha$-$Fe_2O_3$ oxygen vacancy sites (R1 and R2).

$O$ (vacancy) $+ O_2 + 2e^- \rightarrow 2O^-$      (1)

$SO_3^{2-} + O^- \rightarrow SO_4^{2-} + e^-$      (2)

Besides, the change of pH may promote the pH-dependent process such as $O_3$, TMI, $NO_X$. Botha et al. (1994) confirmed that the rate of S(IV) oxidation by $O_3$ increases by several orders of magnitude as the pH increases above 5.5. Wang et al. (2016) analyzed that $NO_X$ made an efficient conversion of $SO_2$ to $SO_4^{2-}$ at high pH. And the oxidation rate of $SO_2$ by TMI was the highest at pH = 5-6 proved by Fuzzi, (1978). Hence, in our experiments, the presence of $NH_3$ enhanced the formation of sulfate.

For more clarity, the following statements have been included in the revised manuscript:

**Page 5, Lines 169:**

When ammonia was in excess, sulfate aerosol should be mainly presented as ammonium sulfate (Silvern et al., 2017). The surface Lewis basicity on the mineral increased the amount of condensed water on the secondary aerosols and enhanced the formation of sulfate (Chu et al., 2016). Also, the increase of pH in liquid due to the presence of $NH_3$ may promote the oxidation by TMI. The rate of S(IV) oxidation by TMI was the highest at pH = 5–6 (Fuzzi, 1978). Besides, when $NH_3$ adsorbed on the surfaces of oxides, synergistic adsorption of $SO_2$ may occur as $SO_3^{2-}$ via reactions (9), (10). In the presence of $NH_3$, the process consisting of oxidation of $HSO_3^-$ or $SO_3^{2-}$ to $SO_4^{2-}$ in the presence of oxygen may be involved in the formation of sulfate (Chu et al., 2016). The oxygen vacancies in $\alpha$-$Fe_2O_3$ may also lead to the formation of sulfate on $\alpha$-$Fe_2O_3$ in the presence of $NH_3$ (R11 and R12) (Yang et al., 2016). Herein, the results showed that the presence of $NH_3$ may lead to an increase in the formation of sulfate.

$SO_2 + OH^- \rightarrow HSO_3^{2-}$      (R9)

$SO_2 + 2OH^- \rightarrow SO_3^{2-} + H_2O$      (R10)

$O$ (vacancy) $+ O_2 + 2e^- \rightarrow 2O^-$      (R11)

$SO_3^{2-} + O^- \rightarrow SO_4^{2-} + e^-$      (R12)

The new references are cited here.

**Reference:**

Baltrusaitis, J., Cwiertny, D. M., Grassian, V. H.: Adsorption of sulfur dioxide on hematite and goethite particle surfaces, Phys. Chem. Chem. Phys., 9, 5542–5554, 2007.

Botha, C. F., Hahn, J., Pienaar, J. J., and Vaneldik, R.: Kinetics and mechanism of the oxidation of sulfur(IV) by ozone in aqueous solutions, Atmos. Environ., 28, 3207–3212, 1994.

Fuzzi, S.: Study of iron (III) catalyezd sulphur dioxide oxidation aqueous solution over a wide rang of pH, Atmos. Environ., 12, 1439–1442, 1978.

- The most important concern related to this experiment is what oxidation processes were included in each experimental system. Previous experimental results by Harris et al. showed the S isotope fractionations for gas-phase oxidation (i.e. $SO_2 + OH$) and aqueous oxidations by $O_3$, $H_2O_2$, and $O_2$ catalyzed by TMI. They also reported fractionation in $SO_2$ oxidation on the dust surface. Compared to these results, this manuscript provides S isotopic fractionation for sulfate formation with different conditions, but I do not understand which oxidation processes were occurred in each system. Simply speaking, I do not understand which oxidants worked in each condition. Probably, there were mixed effects of different oxidation processes, which is so confusing. Thus, I do not agree that this experiment can directly be applied for the interpretation of observational data sets.

**Response:** Thank Reviewer#2 for his/her thoughtful suggestions. Indeed, there were mixed effects of different oxidation processes in each condition. In the presence of $NO_X$, $NO_X$ and $O_2$ worked as oxidants. However, $NO_X$ dominating in the reaction resulted in the enrichment of heavy sulfur isotope. In the presence of $O_3$, both $O_3$ and $O_2$ were oxidants. Compared with $NO_X$, the presence of $O_3$ and $O_2$ favored lighter sulfur isotopes. In the presence of $NH_3$, $O_2$ was the only oxidant and $NH_3$ can promote the formation of sulfate. Moreover, this experiment was not directly applied for the interpretation of observational data sets actually. The observational data sets showed that there existed fractionation effects in the process of $SO_2$ oxidation in the atmosphere. Hence, sulfur isotope fractionation can be used to investigate the effects of $NO_X$, $O_3$, and $NH_3$ on the heterogeneous oxidation of $SO_2$ on $\alpha$-$Fe_2O_3$ surface in the lab experiments.

- The 2nd important concern is the conclusions of this study that $NO_X$ played a major in the different heterogeneous oxidation process of $SO_2$, which cannot be lead by these experimental results and interpretation. Particularly, in eq (3), authors hypothesized that sulfate is only formed via three pathways of $SO_2 + NO_X$, $O_3$, and $NH_3$, but this is not true (as mentioned above). Thus, the conclusion lead by this calculation is not appropriate. These comments are almost the same as referee #1 of "how is this possible?".

**Response:** Thank for Reviewer#2's reasonable comments. The experiments conducted in the laboratory were all on the surface of mineral dust. The sulfur isotope compositions were investigated to determine the effect of $NO_X$, $O_3$ and $NH_3$ on the heterogeneous oxidation of $SO_2$ on $\alpha$-$Fe_2O_3$. Hence, the contribution of TMI oxidation was not taken into account since the reactions were all carried out on the surface of the mineral and we only calculated the contributions of the three gases of $NO_X$, $O_3$, and $NH_3$ on the mineral dust in our experiments. Besides, we used linear fitting to solve the equation (3). According to the sulfur isotopic fractionation and Eq (3), we calculated that the contribution of $NO_X$ accounted for a larger proportion. So we concluded that $NO_X$ played a major in the different

heterogeneous oxidation processes of SO$_2$ on the surface of α-Fe$_2$O$_3$. The following statements have been included in the revised manuscript.

**Page 1, Lines 12:**

    To understand the mechanism of sulfate formations, the characteristics of sulfur isotope composition were determined during different heterogeneous oxidation reactions of sulfur dioxide on the surface of α-Fe$_2$O$_3$.

**Page 1, Lines 20:**

    Given the isotope mass balance, the overall δ$^{34}$S$_{sulfate}$ approached the δ$^{34}$S$_{emission}$ as oxidation of SO$_2$ progressed, suggesting that NO$_X$ played a major rather than a sole role in the different heterogeneous oxidation processes of SO$_2$ on the mineral dust surface.

**Page 3, Lines 71:**

    Herein, for the first time, the several SO$_2$ oxidation processes with the different chemical conditions (NO$_X$, O$_3$, and NH$_3$) are conducted on the hematite surface in the laboratory to gain insight into the sulfur isotope fractionation.

**Page 6, Lines 194:**

    To make clear the relative contribution of NO$_X$, O$_3$, and NH$_3$ during the heterogeneous oxidation of SO$_2$ on the surface of mineral dust, we investigated the isotope fractionation of sulfate.

**Page 7, Lines 214:**

    Considering the isotope mass balance, the overall ε$_{obs}$ value (+1.3 ‰) fell in between NOx, O$_3$, and NH$_3$ values and approached δ$^{34}$S$_{emission}$ as oxidation of SO$_2$ progressed, indicating that NO$_X$, O$_3$, and NH$_3$ all had a certain influence on the heterogeneous oxidation of SO$_2$ on the surface of α-Fe$_2$O$_3$.

**Page 7, Lines 221:**

    The contributions of NO$_X$, O$_3$, and NH$_3$ pathways determined by Eq. 2 and linear regression were 67.5±10 %, 13.3±10 %, 19.2±10 %, respectively.

**Page 7, Lines 224:**

    However, NO$_X$ accounted for a larger proportion in sulfur isotope fractionation due to its tendency to enrich heavier sulfur, suggesting the NO$_X$ pathway has played a more important role during the oxidation of SO$_2$ on the hematite surface.

**Page 7, Lines 231:**

    On the basis of the isotope mass balance, we concluded that NO$_X$ pathway was a dominating but not sole pathway during different heterogeneous oxidation processes of SO$_2$ on α-Fe$_2$O$_3$, which laid the foundation for the research of the mechanism of sulfate formation.

**Response to Reviewer#3' s comments**

- L 104: As shown in Fig. 2, the yield of $SO_4^{2-}$ ranged from 0.0097 to 0.7795 g and the values of δ34S were 2.9–4.8 ‰. I see no sulfate concentration above 0.6 g in figure 2, and no delta values below 3.5 ‰.

**Response:** We are very sorry for not checking the consistency between text and figures. We carefully check the manuscript and revised this sentence as follows.

**Page 4, Lines 114:**
    As shown in Fig. 2, the yield of $SO_4^{2-}$ ranged from 0.0097 to 0.5789 g and the values of $δ^{34}S$ were 3.5–4.8 ‰.

- L129/130: The yield of $SO_4^{2-}$ ranged from 0.0081 to 0.6712 g with the δ34S values of 1.6–2.9 ‰ (Fig. 3). I see no sulfate concentration above 0.6 g in figure 3.

**Response:** We are very sorry for the cursoriness. The sentence was checked in the manuscript and we revised this sentence as follows.

**Page 4, Lines 138:**
    The yield of $SO_4^{2-}$ ranged from 0.0081 to 0.5572 g with the $δ^{34}S$ values of 1.6–2.8 ‰ (Fig. 3).

- Line 157: yield of $SO_4^{2-}$ ranged from 0.0237 to 0.9469 g with the δ34 S values of 0.8–4.3 ‰ (Fig. 4). I neither see the very small or very high sulfate values mentioned in the text in Fig. 4. The range in figure 4 (1.2-2.5 ‰) is very different from what is written in the text.

**Response:** We are very apologetic for the thoughtlessness. We carefully check the manuscript and revised this sentence as follows.

**Page 5, Lines 167:**
    It can be observed that the yield of $SO_4^{2-}$ ranged from 0.0237 to 0.6527 g with the $δ^{34}S$ values of 1.15–2.48 ‰ (Fig. 4).

- The methods section provides almost no information on experimental conditions and setup, I have to guess how experiments were conducted. I wonder whether there is any relation of the samples collected outside and the lab experiments. I understand from the description that the concentrations of the reactants are in the % range, is this then representative for the atmosphere? Why were the various flow rates used? How are samples collected in the lab experiments? These are only a few questions.

**Response:** We completely agree with the valuable suggestions. The apparatus is equipped with temperature and humidity adjustments, as well as a UV lamp. The sample plate in the apparatus is available to be taken out after the reaction. Under different reaction conditions, gas reactants were added to the apparatus from the gas inlet to react with the sample. Besides, the observational data sets were used for comparison with results in experiments. The observational data sets showed that there existed fractionation effects in the process of $SO_2$ oxidation in the atmosphere. Hence, sulfur isotope

fractionation can be used to investigate the effects of $NO_X$, $O_3$, and $NH_3$ on the heterogeneous oxidation of $SO_2$ on $\alpha$-$Fe_2O_3$ surface in the lab experiments. Moreover, the concentrations of the reactants in the experiments were not representative of the atmosphere. The experiments were performed in limited physical boundaries to explore the mechanism of sulfur isotope fractionation on the microscale. High concentrations of reactants are beneficial for our study to investigate the effects precisely. Furthermore, the various flow rates were used to change the concentration of $NO_X$, $O_3$, and $NH_3$. The change of concentration ($NO_X$, $O_3$, and $NH_3$) is useful for us to study the effects of gas concentration on the heterogeneous oxidation of $SO_2$ on hematite. Additionally, the sample ($\alpha$-$Fe_2O_3$) in the lab experiments was put on a plate. After the reaction, the sample can be collected when taking out the plate.

**Page 3, Lines 86:**

A plate with evenly dispersed $\alpha$-$Fe_2O_3$ powder was loaded into the experimental apparatus. The sample plate in the apparatus is available to be taken out after the reaction, and then the samples in the lab experiments can be collected. The apparatus is equipped with temperature and humidity adjustments, as well as a UV lamp. On the basis of $SO_2$-Ar, different proportions of $NO_X$ ($O_3$ or $NH_3$) were added to the reactor from the gas inlet, combined with/without $O_2$ and/or light.

**Page 4, Lines 109:**

The variable $\delta^{34}S_{aerosol}$ values were attributed to the fractionation effects during the process of $SO_2$ oxidation (Li et al., 2020). Hence, sulfur isotope fractionation can be used to investigate the effects of $NO_X$, $O_3$, and $NH_3$ on the heterogeneous oxidation of $SO_2$ on $\alpha$-$Fe_2O_3$ surface in the lab experiments.

- In the discussion, the authors completely mix the interpretation of their results with what was found in previous studies and it is not possible to clearly understand when they are reporting a result from a previous study and when they put the results of their experiments in light of previous findings. I think this paper would require a clear separation of results and discussion. And then still a careful distinction on what is new from the present paper and what has been found before.

**Response:** Thank Reviewer#3 for his/her thoughtful suggestions. We carefully check the manuscript and clearly separate the results in our experiments and results from previous studies. And we carefully refine our own what is new in our results. For more clarity, the following statements have been included in the revised manuscript:

**Page 4, Lines 119:**

In addition, the results in our study supported this view found by Cheng et al. (2016) that the increase in the amount of $NO_X$ was another key factor that led to the acceleration of sulfate formation.

**Page 4, Lines 134:**

Therefore, under dark conditions, the sulfur isotope value was mainly affected by oxygen and the catalytic action of Fe(III), resulting in the enrichment of lighter sulfur isotopes.

**Page 5, Lines 163:**

It is possible that the uptake and decomposition of ozone under irradiation in our study would increase the basicity of the surface, which Hanisch et al. (2003) found to increase enrichment of heavy

sulfur isotopes and is consistent with the increased enrichment in our irradiated ozone experiments.

**Page 6, Lines 184:**

Owing to the increase of alkalinity by producing $OH^-$ from hydrolysis of $NH_3$ (Jiang et al., 2017), $NH_3$ dominated in the reaction under only-light, leading to an increase of $\delta^{34}S$ values in our study.

- Quantitatively, the epsilon value of the overall process (1.3 ‰ line 192), which enables source partitioning, falls completely from heaven. Where does it come from? Does the evaluation include the assumption from the literature that the source isotopic composition is 2.7 ‰ ± 2 ‰ (line 184)? That would result in a huge error. In the present version of the paper, Figure 5 shows some synthetic Rayleigh fractionation curves, but I see no relation to the data, other than the unexplained value of 1.3 ‰.

**Response:** Thank Reviewer#3 a lot for suggestions. We also performed the experiments of mixing three gases ($NO_X$, $O_3$, and $NH_3$) in the laboratory. The epsilon value of 1.3 ‰ was the average value in this experiment. Besides, the evaluation did not include the assumption from the literature that the source isotopic composition is 2.7 ‰ ± 2 ‰. The value of +2.7±2.0 ‰ (Li et al., 2020) was just used as the $\delta^{34}S_{emission}$ value during our sampling period at the field site. It can be observed from Fig. 1 that as oxidation of $SO_2$ progressed, the $\delta^{34}S$ values of aerosol decreased, making them approach $\delta^{34}S_{emission}$. This was attributed to the sulfur isotope mass balance. What's more, the data in Figure 5 were relevant. The $\varepsilon_{NOx}$ value of +2.1 ‰ was calculated based on the $\delta^{34}S$ values in the presence of $NO_X$. The $\varepsilon_{O_3}$ = +2.1 ‰ was calculated based on the $\delta^{34}S$ values in the presence of $O_3$. And -0.6 ‰ was calculated from sulfur isotope values in the presence of $NH_3$. The value of 1.3 ‰ was calculated based on the $\delta^{34}S$ values in the presence of $NO_X$, $O_3$, and $NH_3$. Considering the isotope mass balance, the overall $\varepsilon_{obs}$ value (+1.3 ‰) fell in between $NO_X$, $O_3$, and $NH_3$ values. We also added the point data in the experiments of mixing three gases to figure 5 and deleted the $\delta^{34}S_{emission}$ (+2.7±2 ‰) in figure 5.

**Page 6, Lines 201:**

On the basis of the literature, +2.7±2.0 ‰ is used as the $\delta^{34}S_{emission}$ value during our sampling period at the field site (Li et al., 2020). It can be observed from Fig. 1 that as oxidation of $SO_2$ progressed, the $\delta^{34}S$ values of aerosol decreased, making them approach $\delta^{34}S_{emission}$, which was attributed to sulfur isotope mass balance.

**Page 6, Lines 211:**

The $\varepsilon_{obs}$ values as a result of mixing oxidation pathways of $SO_2$ + $NO_X$ + $O_3$ + $NH_3$ were +1.3±1.4 ‰ measured in our laboratory experiments of mixing three gases ($NO_X$, $O_3$, and $NH_3$) (Fig. 6).

**Page 13, Lines 429-430:**

[Figure]

**Figure 1:** $\delta^{34}S_{aerosol}$ (black), calculated sulfur oxidation ratio (SOR, green) throughout the sample period. The grey bar indicates the $\delta^{34}S_{emission}$ (+2.7±2 ‰) in Nanjing.

**Page 15, Lines 437-440:**

[Figure]

**Figure 5:** Rayleigh distillation model of sulfate production. The red circles are the measured $\delta^{34}S_{obs}$ in this study. Dashed lines with shaded areas are $\delta^{34}S_{sulfate}$ values: the red line indicates the $\delta^{34}S_{sulfate}$ (+4.5±0.5 ‰) when $SO_2$ is oxidized solely by $NO_X$, the blue line indicates the $\delta^{34}S_{sulfate}$ (+2.2±0.3 ‰) when $SO_2$ is oxidized solely by $O_3$, the green line indicates the $\delta^{34}S_{sulfate}$ (+1.6±1.3 ‰) when $SO_2$ is oxidized solely by $NH_3$, and the dark grey line is the $\varepsilon_{obs}$ of +1.3±1.4 ‰. The shaded regions around each line mean the measurement error

- All the data shown in Figure 1-4 should have associated error bars.

**Response:** Thank you for the suggestion. Each point in the figures was based on the single measurement in our experiments. Thus, as Reviewer#1 suggested, we added the perpendicular lines to indicate the accuracy of the measurements.

**Page 13-14, Lines 431-436:**

[Figure]

**Figure 2:** Effect of the mixed gas of SO₂ and NOX on (a) sulfate production and (b) sulfur isotope value under different reaction conditions. The perpendicular lines on the points indicate the accuracy of the measurements.

[Figure]

**Figure 3:** Effect of the mixed gas of SO₂ and O₃ on (a) sulfate production and (b) sulfur isotope values under different reaction conditions. The perpendicular lines on the points indicate the accuracy of the measurements.

[Figure]

**Figure 4:** Effect of the mixed gas of SO₂ and NH₃ on (a) sulfate production and (b) sulfur isotope values under different reaction conditions.